# Generalization bounds for Kernel Canonical Correlation Analysis

**Enayat Ullah** *enayat@jhu.edu*
*Department of Computer Science*
*Johns Hopkins University*

**Raman Arora** *arora@cs.jhu.edu*
*Department of Computer Science*
*Johns Hopkins University*

**Reviewed on OpenReview:** *https://openreview.net/forum?id=KwWKB9Bqam*

## Abstract

We study the problem of multiview representation learning using kernel canonical correlation analysis (KCCA) and establish non-asymptotic bounds on generalization error for regularized empirical risk minimization. In particular, we give fine-grained high-probability bounds on generalization error ranging from $O(n^{-1/6})$ to $O(n^{-1/5})$ depending on underlying distributional properties, where $n$ is the number of data samples. For the special case of finite-dimensional Hilbert spaces (such as linear CCA), our rates improve, ranging from $O(n^{-1/2})$ to $O(n^{-1})$. Finally, our results generalize to the problem of functional canonical correlation analysis over abstract Hilbert spaces.

## 1 Introduction

Canonical correlation analysis (CCA) is a popular technique for multiview representation learning and statistical data analysis. Given a pair of random vectors, CCA finds maximally correlated linear components of the two vectors (Hotelling, 1936). CCA-based methods have recently been shown to improve unsupervised learning of low-dimensional representations of data when multiple "views" of data are available (Vinokourov et al., 2003; Hardoon et al., 2004; Arora and Livescu, 2013). The different views often contain complementary information, and CCA-based multiview representation learning methods can take advantage of this information to learn features that are useful for understanding the structure of the data and that is beneficial for downstream tasks.

Various nonlinear extensions of these multiview learning techniques have also been proposed including kernel CCA (Lai and Fyfe, 2000; Akaho, 2001; Hardoon et al., 2004; Fukumizu et al., 2007) based on positive definite kernels wherein data are represented as functions in associated reproducing kernel Hilbert spaces (RKHS), and deep neural network based extensions, e.g., deep CCA (Andrew et al., 2013)

While CCA and its nonlinear extensions have enjoyed tremendous empirical success, the theoretical understanding of the approaches to solving these problems has been somewhat limited. For example, only recently, were we (as a community) able to give statistical and computational complexity bounds for CCA as a stochastic optimization problem (aka a learning problem) (Allen-Zhu and Li, 2016; Ge et al., 2016; Arora et al., 2017). In a similar spirit of understanding the data analysis techniques as learning problems, in this paper, we look at Kernel CCA and focus on understanding the generalization properties.

However, moving from subspace learning (i.e., linear representations, e.g., using CCA) to learning representations in an RKHS has additional theoretical challenges associated with it. It is then natural to rely on kernel duality, i.e., the representer theorem to reduce the empirical risk minimization (ERM) problem to a finite dimensional optimization problem. Using kernel duality to formulate Kernel CCA was first studied by Lai and Fyfe (2000), Akaho (2001), Melzer et al. (2001) and Bach and Jordan (2002).

In this work, we are interested in understanding the statistical properties of the regularized empirical risk minimizer (defined formally in the subsequent sections) using excess *generalization error* as the error criterion. Informally, excess generalization error of an estimator is the excess error, incurred in objective (or cost), compared to the best, with respect to the underlying data distribution (see Section 3 for a precise formula). This problem has been studied in prior works of (Fukumizu et al., 2007; Fan and Lian, 2016), however their results are asymptotic (see paragraph "Relation to prior work" for more details). Further, these works have studied Kernel CCA in terms of estimation error (or convergence in parameters), and we emphasize that studying the problem in terms of generalization error (or convergence in objective) is important for the following reasons. (Modern) machine learning is typically posed as *risk minimization* problem where the goal is to find parameters that are good in terms of the objective (aka generalization error or population risk) rather than finding the *true* parameters (under some statistical model). Taking a learning view of the KCCA problem, we therefore measure the quality of the solution in terms of the objective rather than distance from a ground truth (which may or may not be unique). This error criterion has been used in the prior works, such as Arora et al. (2017) and Wang et al. (2016), for (linear) CCA. Hence, the main goal in our work, is to give "fine-grained" non-asymptotic guarantees on excess generalization error of regularized empirical risk minimizer (a widely used estimator) for kernel CCA.

## 1.1 Our Contributions

Our main contributions are as follows.

1. We pose kernel CCA as a learning problem and give upper bounds on excess generalization error of the regularized Empirical Risk Minimizer (ERM). Our results hold for the more general problem of functional CCA in abstract Hilbert spaces. To the best of our knowledge, this is the first work which establishes statistical rates of a finite sample estimator for functional CCA. As special cases, our results give generalization bounds for Kernel CCA and linear CCA, and for both of these special cases, we establish novel results compared to previous work (see below).

2. Under standard assumptions (see Assumption 1), we obtain non-asymptotic bounds on excess generalization error of regularized ERM for kernel CCA, which are between $O(n^{-1/6})$ to $O(n^{-1/5})$ depending on  properties of the underlying distribution, where $n$ is the number of data points (see Theorem 1). In contrast, previous works only yield asymptotic guarantees. In the setting when the Reproducing Kernel Hilbert Spaces (RKHS) are finite dimensional, we obtain faster rates ranging from $O(n^{-1/2})$ to $O(n^{-1})$ (see Corollary 3). In the special case of linear CCA, our optimistic rate (i.e. $O(n^{-1})$) is better that the previous result of Gao et al. (2017) and the worse case rate is better in the regime where eigengap of covariance matrix at $k$ is $o(1/\sqrt{n})$ (see Section 4 for details).

3. Our analysis provides insights on the role that regularization parameter plays towards trading off approximation error (bias) and estimation error (variance) and in ensuring statistical consistency of the estimator. In particular, in our bounds, the regularization parameter can decay as $\omega(n^{-1/2})$ and ensure statistical consistency of the estimator – see paragraph "Regularization parameter" in Section 4 for details.

Our proof strategy is to decouple the estimation and approximation errors in the learning problem, bound them separately and balance the tradeoff (between them). To bound the estimation error, the primary tool we use is local Rademacher complexity analysis (Bartlett et al., 2002), which allows us to get a spectrum of rates, from worst case to optimistic (depending on how "easy" the problem is). In the context of kernel methods, these techniques have been applied to give improved rates for kernel principal component analysis (Blanchard et al., 2007), support vector machines (SVMs) with random Fourier features (Gilbert et al., 2018) and other kernel learning problems (Mendelson, 2003; Cortes et al., 2013; Ullah et al., 2018). Please see Section 5 for a detailed proof sketch.

**Relation to prior work.** Herein, we informally discuss how our work compares with prior results. We refer the reader to paragraph "Comparison with prior works", in Section 4 for more details. Previous work has studied the the statistical properties of Kernel CCA through the lens of statistical estimation of

| Problem | Error criterion | Convergence rate | Reference |
|---------|-----------------|------------------|-----------|
| Kernel CCA | Parameter | $o_p(1)$ | Fukumizu et al. (2007) |
| Kernel CCA | Parameter | $O_p(n^{-\alpha/(\alpha+1)})^\dagger$ | Fan and Lian (2016) |
| Kernel CCA | Objective | $O(n^{-1/6})$ to $O(n^{-1/5})$ | Ours (Corollary 2) |
| (Linear) CCA | Objective | $O((\text{gap}^2 n)^{-1})^\ddagger$ | Gao et al. (2017) |
| (Linear) CCA | Objective | $O((\text{gap}\sqrt{n})^{-1})$ to $O((\text{gap } n)^{-1})^\ddagger$ | Ours (Corollary 3). |

Table 1: Summarizing our results in context of relevant prior works. In the table, "Parameter" and "Objective" stand for convergence in parameter and objective respectively (see Section 4 for details). †: obtained under additional assumption on eigenvalues of covariance operators – see Eqn. (3), and $\alpha = \min(\alpha_{\mathcal{X}}, \alpha_{\mathcal{Y}})$ therein. ‡ : gap $= \lambda_1(\text{C}) - \lambda_2(\text{C})$ is the eigengap.

parameters. (i.e by bounding *estimation error*). In particular, the works that are most related to ours are that of of Fukumizu et al. (2007) and Fan and Lian (2016). Under standard assumptions, Fukumizu et al. (2007) established statistical consistency of the regularized ERM solution if the regularization parameter $\lambda = \omega(n^{-1/3})$. More recently, Fan and Lian (2016) established minimax statistical rates for Kernel CCA under additional assumptions on the problem. The guarantees in both Fukumizu et al. (2007) and Fan and Lian (2016) are asymptotic. Fukumizu et al. (2007) show that as number of samples $n \to \infty$, the *estimation error* goes to 0, in probability. The work of Fan and Lian (2016) gave rates but these are also in the *convergence in probability* sense. To elaborate, they consider the event that the *estimation error* random variable grows faster that certain sequence in $n$, and show that probability of this event is limiting to 0. These notions do not give any quantitative finite sample guarantees, and are even weaker than convergence in mean. On the other hand, our guarantees are non-asymptotic - the bounds hold with probability, say at least $1 - \delta$, over the randomness in data for any sample size; and importantly the sample complexity bounds only has $\text{poly}(\log(1/\delta))$ dependence in the failure parameter $\delta$ - what are known as "high confidence" guarantees.

In Table 1, we give a summary of our results in context of prior works.

**Organization.** The rest of the paper is organized as follows. We give mathematical preliminaries in Section 2. In Section 3, we present functional and kernel CCA as learning problems, emphasizing the role of kernel duality and regularization. In Section 4, we present our main result and discuss various implications. Finally, in Section 5, we conclude a brief sketch of the proof.

## 2 Preliminaries

In this section, we quickly review some mathematical preliminaries in functional analysis; a didactic treatment of random variables in Hilbert spaces, reproducing kernel Hilbert spaces and Local Rademacher complexity is presented in Appendix A.

Let $(\mathcal{H}_{\mathcal{X}}, \mathcal{F}_{\mathcal{X}}, \rho_{\mathcal{X}})$ and $(\mathcal{H}_{\mathcal{Y}}, \mathcal{F}_{\mathcal{Y}}, \rho_{\mathcal{Y}})$ be two measurable Hilbert spaces where $\mathcal{H}_{\mathcal{X}}, \mathcal{H}_{\mathcal{Y}}$ are separable spaces, $\mathcal{F}_{\mathcal{X}}, \mathcal{F}_{\mathcal{Y}}$ are $\sigma$-fields and $\rho_{\mathcal{X}}, \rho_{\mathcal{Y}}$ are probability measures. Let $\left\{ e_i^{\mathcal{X}} \right\}_{i \in \mathbb{N}}$ and $\left\{ e_i^{\mathcal{Y}} \right\}_{i \in \mathbb{N}}$ be an orthonormal basis for $\mathcal{H}_{\mathcal{X}}$ and $\mathcal{H}_{\mathcal{Y}}$ respectively. Let $h_1, h_1' \in \mathcal{H}_{\mathcal{X}}$ and $h_2, h_2' \in \mathcal{H}_{\mathcal{Y}}$. We use $\langle h_1, h_1' \rangle_{\rho_{\mathcal{X}}}$ or $\langle h_1, h_1' \rangle_{\mathcal{H}_{\mathcal{X}}}$, as per convenience, to denote the inner product between two elements. Similarly we use $\|h_1\|_{\rho_{\mathcal{X}}}$ or $\|h_1\|_{\mathcal{H}_{\mathcal{X}}}$ for norms.

An operator $\text{D} : \mathcal{H}_{\mathcal{Y}} \to \mathcal{H}_{\mathcal{X}}$ is bounded if its operator norm $\|\text{D}\|$, defined as $\|\text{D}\| := \sup\{\|\text{Dh}\|_{\mathcal{H}_{\mathcal{X}}}, \text{h} \in \mathcal{H}_{\mathcal{Y}}, \|\text{h}\|_{\mathcal{H}_{\mathcal{Y}}} \le 1\} < \infty$. The outer product $h_1 \otimes_{L(\mathcal{H}_{\mathcal{Y}}, \mathcal{H}_{\mathcal{X}})} h_2$ is an operator from $\mathcal{H}_{\mathcal{Y}}$ to $\mathcal{H}_{\mathcal{X}}$, which acts as $(h_1 \otimes_{L(\mathcal{H}_{\mathcal{Y}}, \mathcal{H}_{\mathcal{X}})} h_2)h = \langle h_2, h \rangle_{\mathcal{H}_{\mathcal{Y}}} h_1$ for $h \in \mathcal{H}_{\mathcal{Y}}$. The adjoint operator of D, denoted as $\text{D}^* : \mathcal{H}_{\mathcal{X}} \to \mathcal{H}_{\mathcal{Y}}$, is defined as $\langle h_1, \text{D}h_2 \rangle_{\mathcal{H}_{\mathcal{X}}} = \langle \text{D}^* h_1, h_2 \rangle_{\mathcal{H}_{\mathcal{Y}}}$.

A bounded operator is self-adjoint if $\text{D}^* = \text{D}$. An operator $\text{D} : \mathcal{H}_{\mathcal{Y}} \to \mathcal{H}_{\mathcal{X}}$ is Hilbert-Schmidt if its Hilbert-Schmidt norm, denoted as $\|\text{D}\|_{L(\mathcal{H}_{\mathcal{Y}}, \mathcal{H}_{\mathcal{X}})} := \sum_{i \in \mathbb{N}} \left\| \text{De}_i^{\mathcal{Y}} \right\|_{\mathcal{H}_{\mathcal{X}}}^2 = \sum_{i,j \in \mathbb{N}} \left\langle \text{De}_i^{\mathcal{Y}}, e_j^{\mathcal{X}} \right\rangle_{\mathcal{H}_{\mathcal{X}}} < \infty$. We use $L(\mathcal{H}_{\mathcal{Y}}, \mathcal{H}_{\mathcal{X}})$ to denote all Hilbert-Schmidt operators from $\mathcal{H}_{\mathcal{Y}}$ to $\mathcal{H}_{\mathcal{X}}$. For the sake of brevity, we use $L(\mathcal{H}_{\mathcal{X}})$ to denote Hilbert-Schmidt operators from $\mathcal{H}_{\mathcal{X}}$ to $\mathcal{H}_{\mathcal{X}}$. An operator $\mathcal{D} : \mathcal{H}_{\mathcal{X}} \to \mathcal{H}_{\mathcal{Y}}$ is compact if the image of

every bounded subset of $\mathcal{H}_\mathcal{X}$ is a relatively compact subset of $\mathcal{H}_\mathcal{Y}$. A compact operator $D : \mathcal{H}_\mathcal{X} \to \mathcal{H}_\mathcal{X}$ is trace-class if $\|D\|_{L^1(\mathcal{H})} := \sum_{i \geq 1} \langle (DD^*)^{1/2} e_i^1, e_i^1 \rangle_{\mathcal{H}_\mathcal{X}} < \infty$, where $\|D\|_{L^1(\mathcal{H}_\mathcal{X})}$ denotes the nuclear norm of D. An operator $D : \mathcal{H}_\mathcal{X} \to \mathcal{H}_\mathcal{X}$ is positive if $\forall\, f \in \mathcal{H}_\mathcal{X}, \langle f, Df \rangle_{\mathcal{H}_\mathcal{X}} \geq 0$. The identity operator $I_\mathcal{X} : \mathcal{H}_\mathcal{X} \to \mathcal{H}_\mathcal{X}$ is defined as $I_\mathcal{X} f = f \,\forall\, f \in \mathcal{H}_\mathcal{X}$.

**Notation.** We use capital Roman letters (e.g., A) to denote matrices and operators, small Roman letters (e.g., a) for vectors and small letters (e.g., $a$) for scalars. Operators over the space of Hilbert-Schmidt operators are represented using capital Fraktur letters, e.g., $\mathfrak{A}$. For a Hilbert-Schmidt operator D, $\lambda_i(D)$ denotes its $i^{\text{th}}$ eigenvalue. Similarly, $\sigma_i(D)$ denotes the $i^{\text{th}}$ singular value of D. We use $P_D^k$ to denote the top rank $k$ projection of D; for example, if the Singular Value Decomposition (SVD) of D is $D = \sum_{i \in \mathbb{N}} \lambda_i u_i \otimes v_i$, then $P_D^k = \sum_{i=1}^k u_i \otimes v_i$. We use $I_k \in \mathbb{R}^{k \times k}$ to denote a $k \times k$ identity matrix. Natural numbers are denoted by $\mathbb{N}$; $[n]$ denotes natural numbers from 1 to $n$.

## 3 Problem Setup and Background

We begin by recalling the finite dimensional CCA problem. For paired random vectors, $x \in \mathcal{X} \subseteq \mathbb{R}^{d_\mathcal{X}}$ and $y \in \mathcal{Y} \subseteq \mathbb{R}^{d_\mathcal{Y}}$, with some unknown joint distribution $\rho$, Canonical Correlation Analysis (CCA) can be posed as the following problem.

$$\underset{U \in \mathbb{R}^{d_\mathcal{X} \times k}, V \in \mathbb{R}^{d_\mathcal{Y} \times k}}{\text{maximize}} \left\langle UV^\top, \mathbb{E}_{x,y} \left[ xy^\top \right] \right\rangle \text{ such that } U^\top \mathbb{E}_x \left[ xx^\top \right] U = I_k, V^\top \mathbb{E}_y \left[ yy^\top \right] V = I_k,$$

where $\langle A, B \rangle = \text{Trace}(A^\top B)$ is the standard inner product on matrices.

**Functional CCA.** We can generalize the above formulation to abstract Hilbert spaces. In particular, when x and y are random variables in Hilbert spaces $\mathcal{H}_\mathcal{X}$ and $\mathcal{H}_\mathcal{Y}$, respectively, the functional CCA problem can be formulated as,

$$\underset{U \in L(\mathcal{H}_\mathcal{X}, \mathbb{R}^k), V \in L(\mathcal{H}_\mathcal{Y}, \mathbb{R}^k)}{\text{maximize}} \langle UV^*, C_{\mathcal{X}\mathcal{Y}} \rangle_{L(\mathcal{H}_\mathcal{Y}, \mathcal{H}_\mathcal{X})} \text{ such that } U^* C_\mathcal{X} U = I_k, V^* C_\mathcal{Y} V = I_k,$$

where $C_\mathcal{X} = \mathbb{E} \left[ x \otimes_{L(\mathcal{H}_\mathcal{X})} x \right]$ and $C_\mathcal{Y} = \mathbb{E} \left[ y \otimes_{L(\mathcal{H}_\mathcal{Y})} y \right]$ are auto-covariance operators, and $C_{\mathcal{X}\mathcal{Y}} = \mathbb{E} \left[ x \otimes_{L(\mathcal{H}_\mathcal{Y}, \mathcal{H}_\mathcal{X})} y \right]$ is a cross-covariance operator (we refer the reader to Appendix A for a definition).

**Kernel CCA.** Nonlinear CCA extends the problem of CCA in to a high dimensional feature space using nonlinear feature maps. Kernel CCA is one popular variant of nonlinear CCA where the feature maps are implicit in the kernel functions. It can be viewed as a special case of functional CCA with RKHS $\mathcal{H}_\mathcal{X}$ and $\mathcal{H}_\mathcal{Y}$ over $\mathcal{X}$ and $\mathcal{Y}$ associated with kernel functions $k_\mathcal{X}$ and $k_\mathcal{Y}$, respectively:

$$\underset{U \in L(\mathcal{H}_\mathcal{X}, \mathbb{R}^k), V \in L(\mathcal{H}_\mathcal{Y}, \mathbb{R}^k)}{\text{maximize}} \langle UV^*, C_{\mathcal{X}\mathcal{Y}} \rangle_{L(\mathcal{H}_\mathcal{Y}, \mathcal{H}_\mathcal{X})} \text{ such that } U^* C_\mathcal{X} U = I_k, V^* C_\mathcal{Y} V = I_k,$$

where $C_\mathcal{X} = \mathbb{E} \left[ k_\mathcal{X}(x, \cdot) \otimes_{L(\mathcal{H}_\mathcal{X})} k_\mathcal{X}(x, \cdot) \right], C_\mathcal{Y} = \mathbb{E} \left[ k_\mathcal{Y}(y, \cdot) \otimes_{L(\mathcal{H}_\mathcal{Y})} k_\mathcal{Y}(y, \cdot) \right]$ are auto-covariance operators, and $C_{\mathcal{X}\mathcal{Y}} = \mathbb{E} \left[ k_\mathcal{X}(x, \cdot) \otimes_{L(\mathcal{H}_\mathcal{Y}, \mathcal{H}_\mathcal{X})} k_\mathcal{Y}(y, \cdot) \right]$ is the cross-covariance operator.

**Basis transformation.** An alternative equivalent formulation of CCA is obtained by rotating the components in the canonical basis. We then get the following formulation.

$$\underset{U \in L(\mathcal{H}_\mathcal{X}, \mathbb{R}^k), V \in L(\mathcal{H}_\mathcal{Y}, \mathbb{R}^k)}{\text{maximize}} \left\langle UV^*, C_\mathcal{X}^{-1/2} C_{\mathcal{X}\mathcal{Y}} C_\mathcal{Y}^{-1/2} \right\rangle_{L(\mathcal{H}_\mathcal{Y}, \mathcal{H}_\mathcal{X})} \text{ such that } U^* U = I_k, V^* V = I_k.$$

At first, the reformulation above seems to require that the auto-covariance operators are invertible. In the context of kernel CCA, if the feature space corresponding to the kernel function is finite dimensional (for example, when using polynomial kernels), then invertibility can hold. However, it does not hold in general, for example when using a Gaussian kernel, the auto-covariance operators can no longer be trace-class. This is a problem since the standard assumption of random variables being bounded implies the corresponding

auto-covariance operators are trace class and therefore we have a contradiction. However, note that we can instead write the above equivalently as,

$$\underset{U \in L(\mathcal{H}_\mathcal{X}, \mathbb{R}^k), V \in L(\mathcal{H}_\mathcal{Y}, \mathbb{R}^k)}{\text{maximize}} \langle UV^*, C \rangle_{L(\mathcal{H}_\mathcal{Y}, \mathcal{H}_\mathcal{X})} \quad \text{s.t.} \quad C_\mathcal{X}^{1/2} C C_\mathcal{Y}^{1/2} = C_{\mathcal{X}\mathcal{Y}}, U^*U = I_k, V^*V = I_k. \tag{1}$$

The existence and uniqueness of such an operator C, bounded as $\|C\| \leq 1$, is established in Baker (1973, Theorem 1) and also discussed in Section 2.2 of Fukumizu et al. (2007). As in Fukumizu et al. (2007), we assume that C is compact and abuse the notation to write $C = C_\mathcal{X}^{-1/2} C_{\mathcal{X}\mathcal{Y}} C_\mathcal{Y}^{-1/2}$ even when these are not invertible.

We now discuss a final assumption. Note that since $C_\mathcal{X}$ and $C_\mathcal{Y}$ are self-adjoint, a spectral decomposition of $C_\mathcal{X}$ and $C_\mathcal{Y}$ exists. Let $C_\mathcal{X} := \sum_{i=1}^\infty \lambda_i(C_\mathcal{X}) \phi_i^\mathcal{X} \otimes_{L(\mathcal{H}_\mathcal{X})} \phi_i^\mathcal{X}$ and $C_\mathcal{Y} := \sum_{i=1}^\infty \lambda_i(C_\mathcal{Y}) \phi_i^\mathcal{Y} \otimes_{L(\mathcal{H}_\mathcal{Y})} \phi_i^\mathcal{Y}$ where $\left\{ \phi_i^\mathcal{X} \right\}_i$ and $\left\{ \phi_i^\mathcal{Y} \right\}_i$ are the eigenfunctions of $C_\mathcal{X}$ and $C_\mathcal{Y}$ respectively. We define $\gamma_{ij} := \mathbb{E}_{x,y} \left[ \langle \phi_i^\mathcal{X} \otimes_{L(\mathcal{H}_\mathcal{Y}, \mathcal{H}_\mathcal{X})} \phi_i^\mathcal{Y}, C_{\mathcal{X}\mathcal{Y}} \rangle_{L(\mathcal{H}_\mathcal{Y}, \mathcal{H}_\mathcal{X})} \right]$. We assume that $\max_{i,j} \left| \frac{\gamma_{ij}}{\lambda_i(C_\mathcal{X})\sqrt{\lambda_i(C_\mathcal{Y})}} \right| \leq 1$ and $\max_{i,j} \left| \frac{\gamma_{ij}}{\sqrt{\lambda_i(C_\mathcal{X})\lambda_i(C_\mathcal{Y})}} \right| \leq 1$. With abuse of notation, this means that the operators $C_\mathcal{X}^{-1} C_{\mathcal{X}\mathcal{Y}} C_\mathcal{Y}^{-1/2}$ and $C_\mathcal{X}^{-1/2} C_{\mathcal{X}\mathcal{Y}} C_\mathcal{Y}^{-1}$ are bounded, and their operator norms bounded by 1. These assumptions have appeared in previous works Fukumizu et al. (2007) and Fan and Lian (2016) and ensures existence of a solution to the kernel CCA problem.

Observe that the solution to the CCA problem in Eqn. (1) is given by the singular value decomposition (SVD) of C. In particular, let $u_1^C, u_2^C, \dots, u_k^C$ and $v_1^C, v_2^C, \dots, v_k^C$ be the top-$k$ left and right singular functions of C, respectively. We define $U_C : \mathbb{R}^k \to \mathcal{H}_\mathcal{X}$ such that $U_C b = \sum_{i=1}^k b_i u_i^C$, where $b \in \mathbb{R}^k$ ; and similarly $V_C$. The solution to the CCA problem in Eqn. (1) is $U_C$ and $V_C$.

**Empirical Risk Minimization (ERM).** We now discuss the learning problem and a finite sample estimator. Let $\{(x_i, y_i)\}_{i=1}^n$ be $n$ data points drawn i.i.d. from $\rho$. We first define empirical counterparts of auto covariance and cross-covariance operators. We define $C_\mathcal{X}^n : \mathcal{H}_\mathcal{X} \to \mathcal{H}_\mathcal{X}$ and $C_\mathcal{Y}^n : \mathcal{H}_\mathcal{Y} \to \mathcal{H}_\mathcal{Y}$ as $C_\mathcal{X}^n = \frac{1}{n} \sum_{i=1}^n x_i \otimes_{L(\mathcal{H}_\mathcal{X})} x_i$ and $C_\mathcal{Y}^n = \frac{1}{n} \sum_{i=1}^n y_i \otimes_{L(\mathcal{H}_\mathcal{Y})} y_i$, respectively. Similarly, the empirical cross-covariance operator $C_{\mathcal{X}\mathcal{Y}}^n = \frac{1}{n} \sum_{i=1}^n x_i \otimes_{L(\mathcal{H}_\mathcal{Y}, \mathcal{H}_\mathcal{X})} y_i$. We also define $C^n := (C_\mathcal{X}^n)^{-1/2} C_{\mathcal{X}\mathcal{Y}}^n (C_\mathcal{Y}^n)^{-1/2}$. ERM is formulated as,

$$\underset{U \in L(\mathcal{H}_\mathcal{X}, \mathbb{R}^k), V \in L(\mathcal{H}_\mathcal{Y}, \mathbb{R}^k)}{\text{maximize}} \langle UV^*, C^n \rangle_{L(\mathcal{H}_\mathcal{Y}, \mathcal{H}_\mathcal{X})} \text{ such that } \quad U^*U = I_k, V^*V = I_k.$$

The solution to the ERM problem above, analogously, are the singular functions of $C^n$. Let $u_1^n, u_2^n, \dots, u_k^n$ and $v_1^n, v_2^n, \dots, v_k^n$ be the top-$k$ left and right singular functions of $C_n$ respectively. We define $U^n : \mathbb{R}^k \to \mathcal{H}_\mathcal{X}$ such that $U_n b = \sum_{i=1}^k b_i u_i^n$, where $b \in \mathbb{R}^k$, and similarly $V_n$.

**Excess Generalization error.** As motivated in Section 1, we are interested in studying the kernel CCA problem from the point of view of learning, i.e., generalization error. The excess generalization error of an estimator is the excess error incurred, in objective, compared to the best, with respect to the underlying distribution. The excess generalization error of ERM solutions (i.e. $(U_n, V_n)$) is the following,

$$\mathcal{E}(U_n, V_n) = \langle U_C V_C^* - U_n V_n^*, C \rangle_{L(\mathcal{H}_\mathcal{Y}, \mathcal{H}_\mathcal{X})}.$$

**Kernel duality.** Note that even if we establish statistical convergence of the estimator, we cannot talk about computational aspects as these are infinite dimensional objects. However, in the special case of RKHS, we can appeal to kernel duality to guarantee efficient computation. In the context of CCA, by observing that the solution should lie in the span of data, Akaho (2001) and Bach and Jordan (2002) show that the empirical problem is equivalent to solving the following finite dimensional optimization problem.

$$\underset{U \in \mathbb{R}^{n \times k}, V \in \mathbb{R}^{n \times k}}{\text{maximize}} \left\langle UV^\top, \frac{1}{n} K_\mathcal{X} K_\mathcal{Y} \right\rangle \text{ such that } \quad \frac{1}{n} U^\top K_\mathcal{X}^2 U = I_k, \frac{1}{n} V^\top K_\mathcal{Y}^2 V = I_k,$$

where $\mathrm{K}_{\mathcal{X}}$ and $\mathrm{K}_{\mathcal{Y}}$ are $n \times n$ kernel matrices with $(\mathrm{K}_{\mathcal{X}})_{ij} = k_{\mathcal{X}}(\mathrm{x}_i, \mathrm{x}_j)$ and $(\mathrm{K}_{\mathcal{Y}})_{ij} = k_{\mathcal{Y}}(\mathrm{y}_i, \mathrm{y}_j)$. Solving this typically takes $O(n^2 k)$ time and $O(n^2)$ space. However, there are faster approximate alternatives, for example, those based on approximate feature maps (Rahimi and Recht, 2007; Kar and Karnick, 2012), Nyström's method (Drineas and Mahoney, 2005) and sketching (Yang et al., 2015).

## 3.1 Regularization

Empirical risk minimization (ERM) is one of the most popular learning rules. However, without an appropriate inductive bias, e.g., in the form of a regularizer, the ERM solution may fail to generalize. For the ERM problem corresponding to canonical correlation analysis, a natural regularizer is a variant of Tikhonov regularization (Groetsch, 1984), which can be described as follows. Define $\mathrm{C}_{\mathcal{X}}^{n,\lambda_{\mathcal{X}}} := \mathrm{C}_{\mathcal{X}}^n + \lambda_{\mathcal{X}} \mathrm{I}_{\mathcal{X}}$, $\mathrm{C}_{\mathcal{Y}}^{n,\lambda_{\mathcal{Y}}} := \mathrm{C}_{\mathcal{Y}}^n + \lambda_{\mathcal{Y}} \mathrm{I}_{\mathcal{Y}}$ and $\mathrm{C}^{n,\lambda} := (\mathrm{C}_{\mathcal{X}}^{n,\lambda_{\mathcal{X}}})^{-\frac{1}{2}} \mathrm{C}_{\mathcal{X}\mathcal{Y}}^n (\mathrm{C}_{\mathcal{Y}}^{n,\lambda_{\mathcal{Y}}})^{-1/2}$, where $\lambda_{\mathcal{X}}$ and $\lambda_{\mathcal{Y}}$ are regularization parameters, and $\mathrm{I}_{\mathcal{X}}$ and $\mathrm{I}_{\mathcal{Y}}$ are the identity operators over $\mathcal{H}_{\mathcal{X}}$ and $\mathcal{H}_{\mathcal{Y}}$, respectively. The regularized ERM problem for CCA (Hardoon et al., 2004; Vinokourov et al., 2002) is then given as:

$$\underset{\mathrm{U} \in L(\mathcal{H}_{\mathcal{X}}, \mathbb{R}^k), \mathrm{V} \in L(\mathcal{H}_{\mathcal{Y}}, \mathbb{R}^k)}{\text{maximize}} \left\langle \mathrm{UV}^*, \mathrm{C}^{n,\lambda} \right\rangle_{L(\mathcal{H}_{\mathcal{Y}}, \mathcal{H}_{\mathcal{X}})} \text{ such that } \quad \mathrm{U}^*\mathrm{U} = \mathrm{I}_k, \mathrm{V}^*\mathrm{V} = \mathrm{I}_k.$$

The regularization above corresponds to shifting the spectrum of the auto-covariance operators for $\lambda_{\mathcal{X}}, \lambda_{\mathcal{Y}} > 0$, so that all eigenvalues are positive. Intuitively, we can see that this regularizer biases the problem to tradeoff solutions which maximize correlation and are not along directions with small (non-zero) variance. Specifically, since for all practical purposes, we only observe finitely many samples, and so the presence of very small eigenvalues magnifies the spurious correlations observed. It is therefore imperative to minimize the effect of such small eigenvalues.

We emphasize that regularization parameters $\lambda_{\mathcal{X}}$ and $\lambda_{\mathcal{Y}}$ should be viewed as parameters dependent on $n$, as is standard in statistical machine learning. Moreover, in such problems, it is usually expected that the regularization parameter decays *fast* with the number of samples, e.g., as $\mathrm{poly}(1/n)$. In Section 4, we discuss the approximation and estimation error tradeoff and how to set the regularization parameters appropriately to optimize it, to have a small overall excess generalization error. In the case where the auto-covariance operators are already positive definite, i.e., their eigenvalues are lower bounded away from zero, the regularization parameters can be viewed as minimum eigenvalues of the corresponding operators. This gives us the rates for kernel CCA in finite dimensional Hilbert spaces, as presented in Corollary 3.

## 4 Main Results

We first collectively state all necessary and simplifying assumptions, and then state our main theorem. All these assumptions are standard and have appeared in the previous works, and we discussed some of these in the prior section.

**Assumption 1.** *We assume that the Hilbert spaces $\mathcal{H}_{\mathcal{X}}$ and $\mathcal{H}_{\mathcal{Y}}$ are separable, random variables $\mathrm{x}$ and $\mathrm{y}$ are centered and bounded as, $\|\mathrm{x}\|_{\mathcal{H}_{\mathcal{X}}} \leq \beta$ and $\|\mathrm{y}\|_{\mathcal{H}_{\mathcal{Y}}} \leq \beta$ almost surely, that $\mathrm{C}$, defined in Eqn. (1), is compact and the singular values of $\mathrm{C}$ are distinct. We assume that $\left\|\mathrm{C}_{\mathcal{X}}^{-1} \mathrm{C}_{\mathcal{X}\mathcal{Y}} \mathrm{C}_{\mathcal{Y}}^{-1/2}\right\| \leq 1$ and $\left\|\mathrm{C}_{\mathcal{X}}^{-1/2} \mathrm{C}_{\mathcal{X}\mathcal{Y}} \mathrm{C}_{\mathcal{Y}}^{-1}\right\| \leq 1$.*

**Theorem 1** (Functional CCA). *Let $\mathrm{x}$ and $\mathrm{y}$ be random variables in Hilbert spaces $\mathcal{H}_{\mathcal{X}}$ and $\mathcal{H}_{\mathcal{Y}}$, and let $\rho$ be the joint distribution over $\mathcal{H}_{\mathcal{X}} \times \mathcal{H}_{\mathcal{Y}}$, and $\rho_{\mathcal{X}}$ and $\rho_{\mathcal{Y}}$ denote its marginals. Under the conditions of Assumption 1, given data samples $(\mathrm{x}_1, \mathrm{y}_1), (\mathrm{x}_2, \mathrm{y}_2), \ldots, (\mathrm{x}_n, \mathrm{y}_n)$ drawn i.i.d. from $\rho$, with probability at least $1 - \delta$, the excess generalization error of regularized ERM, $\mathrm{U}_{n\lambda}, \mathrm{v}_{n,\lambda}$, is bounded as,*

$$\left\langle \mathrm{U}_{\mathrm{C}} \mathrm{V}_{\mathrm{C}}^* - \mathrm{U}_{n,\lambda} \mathrm{V}_{n,\lambda}^*, \mathrm{C} \right\rangle_{L(\mathcal{H}_{\mathcal{Y}}, \mathcal{H}_{\mathcal{X}})} \leq 4k\left(\sqrt{\lambda_{\mathcal{X}}} + \sqrt{\lambda_{\mathcal{Y}}}\right)\left(1 + \frac{2}{\sigma_k(\mathrm{C})}\right) + \frac{16k\left(\sqrt{\lambda_{\mathcal{X}}} + \sqrt{\lambda_{\mathcal{Y}}}\right)^2}{\sigma_k(\mathrm{C})}$$

$$+ \inf_{h \geq 0}\left\{\frac{12\alpha_\rho h}{\lambda_{\mathcal{X}}\lambda_{\mathcal{Y}}n} + \frac{24\alpha_\rho \sqrt{h}}{\lambda_{\mathcal{X}}\lambda_{\mathcal{Y}}n} + \frac{24}{\sqrt{n}}\sqrt{\frac{k}{\lambda_{\mathcal{X}}\lambda_{\mathcal{Y}}}\sum_{j>h} \lambda_j(\mathfrak{C}')}\right\} + \frac{12\alpha_\rho}{\lambda_{\mathcal{X}}\lambda_{\mathcal{Y}}n} + \frac{22\beta\sqrt{k}\log(1/\delta)}{\left(\sqrt{\lambda_{\mathcal{X}}\lambda_{\mathcal{Y}}}\right)n} + \frac{10\alpha_\rho \log(1/\delta)}{\lambda_{\mathcal{X}}\lambda_{\mathcal{Y}}n}$$

*where* $\alpha_\rho = \mathbb{E}_{(x,y)\sim\rho,(x',y')\sim\rho} \langle x, x' \rangle^2_{\mathcal{H}_\mathcal{X}} \langle y, y' \rangle^2_{\mathcal{H}_\mathcal{Y}} / (\lambda_k(\mathrm{C}) - \lambda_{k+1}(\mathrm{C}))$ *and* $\mathfrak{C}' \in L(L(\mathcal{H}_\mathcal{Y}, \mathcal{H}_\mathcal{X}))$, *defined as* $\mathfrak{C}' :=$
$\mathbb{E}_{x,y}\left[\left(x \otimes_{L(\mathcal{H}_\mathcal{Y}, \mathcal{H}_\mathcal{X})} y\right) \otimes_{L(L(\mathcal{H}_\mathcal{Y}, \mathcal{H}_\mathcal{X}))} \left(x \otimes_{L(\mathcal{H}_\mathcal{Y}, \mathcal{H}_\mathcal{X})} y\right)\right] - \mathrm{C}_{\mathcal{X}\mathcal{Y}} \otimes_{L(L(\mathcal{H}_\mathcal{Y}, \mathcal{H}_\mathcal{X}))} \mathrm{C}_{\mathcal{X}\mathcal{Y}}$.

A few remarks are in order. First, for simplicity, consider the case when $\lambda_\mathcal{X} = \lambda_\mathcal{Y} = \lambda$. If we only consider the dependence on $\lambda$ (which should be set as $1/\mathrm{poly}(n)$), the operator $\mathfrak{C}'$ and $n$, with probability at least $1 - \delta$, we get the following bound,

$$\left\langle \mathrm{U_C V_C^*} - \mathrm{U}_{n,\lambda} \mathrm{V}_{n,\lambda}^*, \mathrm{C} \right\rangle_{L(\mathcal{H}_\mathcal{Y}, \mathcal{H}_\mathcal{X})} \leq O\left( \sqrt{\lambda} + \frac{\log(1/\delta)}{\mathrm{gap}_k(\mathrm{C})\lambda^2 n} + \inf_{h \geq 0} \left\{ \frac{h}{\mathrm{gap}_k(\mathrm{C})\lambda^2 n} + \frac{1}{\lambda\sqrt{n}} \sqrt{\sum_{j>h} \lambda_j(\mathfrak{C}')} \right\} \right) \quad (2)$$

where $\mathrm{gap}_k(\mathrm{C}) = \lambda_k(\mathrm{C}) - \lambda_{k+1}(\mathrm{C})$ is the eigengap of C at $k$, which shows up in our generalization bound. We emphasise that the existence of eigengap at $k$ as well as that the eigenvalues of C are distinct (in Assumption 1) are simplifying assumptions. The analysis goes through even otherwise; however then the dependence on gap at $k$ is replaced by gap at $p^{th}$ singular value where $p > k$. Moreover this dependence on gap, in general, is unavoidable, as evidenced by existing lower bounds in the special case of linear CCA(Gao et al., 2017, Lemma 20).

**Spectrum decay of $\mathfrak{C}'$.** We see that the convergence rate is crucially controlled by the decay of the spectrum of $\mathfrak{C}'$, i.e. the term $\frac{1}{\lambda\sqrt{n}} \sum_{j>h} \lambda_j(\mathfrak{C}')$. In the worst case, it behaves as $O\left(\frac{1}{\lambda\sqrt{n}}\right)$; the best is when $\mathfrak{C}'$ is of finite rank, where it behaves as $O(1/\lambda n)$. Furthermore, if the spectrum has an exponential decay, it is $O(\log n/n)$. We emphasize that this term is the result of the local Rademacher complexity analysis which manifests as a spectrum of convergence rates based on higher-order distributional properties.

**Optimizing the Approximation-Estimation error tradeoff.** We call the first term on the right hand side of the inequality (2) as the *bias* and the second and third term, collectively, as the *variance*. We have that bias is in $O(\sqrt{\lambda})$ but the variance behaves differently depending on the spectrum decay of $\mathfrak{C}'$. In the worst case, the variance is in $O(\lambda^{-1} n^{-1/2})$. Optimizing the tradeoff, we get a rate of $O(n^{-1/6})$ when $\lambda = \Theta(n^{-1/3})$. In the best case, the variance decays as $O(\lambda^{-2} n^{-1})$, which yields an optimistic rate of $O(n^{-1/5})$ when $\lambda = \Theta\left(n^{-5/2}\right)$.

**Kernel CCA.** When Hilbert spaces $\mathcal{H}_\mathcal{X}$ and $\mathcal{H}_\mathcal{Y}$ are RKHS associated with kernel functions $k_\mathcal{X}$ and $k_\mathcal{Y}$ respectively, then Theorem 1 gives statistical rates of convergence of empirical kernel CCA in terms of the objective. In particular, we have the following corollary.

**Corollary 2** (Kernel CCA). *Along with the notations and assumptions of Theorem 1, suppose that $\mathcal{H}_\mathcal{X}$ and $\mathcal{H}_\mathcal{Y}$ are reproducing kernel Hilbert spaces associated with kernel functions $k_\mathcal{X}$ and $k_\mathcal{Y}$, respectively. Then, regularized ERM on $n$ data points outputs $\mathrm{U}_{n,\lambda}\mathrm{V}_{n,\lambda}^*$ which satisfies the following with probability at least $1-\delta$,*

$$\left\langle \mathrm{U_C V_C^*} - \mathrm{U}_{n,\lambda} \mathrm{V}_{n,\lambda}^*, \mathrm{C} \right\rangle_{L(\mathcal{H}_\mathcal{Y}, \mathcal{H}_\mathcal{X})} \leq O\left( \sqrt{\lambda} + \frac{\log(1/\delta)\alpha_\rho}{\lambda^2 n} + \inf_{h \geq 0} \left\{ \frac{h\alpha_\rho}{\lambda^2 n} + \frac{1}{\lambda\sqrt{n}} \sqrt{\sum_{j>h} \lambda_j(\mathfrak{C}')} \right\} \right)$$

*where* $\alpha_\rho = \mathbb{E}_{x,y,x',y'}\left[k_\mathcal{X}(x, x')^2 k_\mathcal{Y}(y, y')^2\right] / (\lambda_k(\mathrm{C}) - \lambda_{k+1}(\mathrm{C}))$.

**Comparison with prior works.** Previous work of Fukumizu et al. (2007) and Fan and Lian (2016) study Kernel CCA, for $k = 1$, in the sense of convergence of parameters, i.e. distance between the true and the estimated solution. In particular, let $(u_1, v_1)$ be the true solution of Eqn. (1) and let$(\hat{u}_1, \hat{v}_1)$ be a candidate solution. The error is then given by $\|\mathrm{C}_\mathcal{X}^{1/2}(u_1 - \hat{u}_1)\|^2_{\mathcal{H}_\mathcal{X}}$ and similarly, $\|\mathrm{C}_\mathcal{Y}^{1/2}(v_1 - \hat{v}_1)\|^2_{\mathcal{H}_\mathcal{Y}}$. We remark that this is a *stronger* notion of convergence and implies convergence in objective. However, (as we will discuss), their implied results are *weaker* than ours. Importantly, both these works only give asymptotic results, in sense of *convergence in probability*; this means that the success probability over the draw of samples is not quantified, but only limiting to 1 as $n \to \infty$. In contrast, our results being non-asymptotic hold for all sample sizes. Finally, our results hold for general $k$ as opposed to the above works which are limited to $k = 1$.

We now discuss both of these works one by one. The goal in Fukumizu et al. (2007) is to establish statistical consistency of the (regularized) ERM. They show that as $n \to \infty$, the error $\|C_{\mathcal{X}}^{1/2}(u_1 - \hat{u}_1)\|_{\mathcal{H}_{\mathcal{X}}}^2 = o_P(1)$, [1] and similarly for error $\|C_{\mathcal{Y}}^{1/2}(v_1 - \hat{v}_1)\|_{\mathcal{H}_{\mathcal{Y}}}^2$. On the other hand, as remarked earlier, we give high-confidence upper bounds on error in objective, as a function of sample size $n$, for any $n$.

The other related work of Fan and Lian (2016) establishes minimax rates for Kernel CCA in the sense of convergence in parameters. However, their results hold under a strict assumption on the decay of eigenvalues of $C_{\mathcal{X}}$ and $C_{\mathcal{Y}}$. In particular, there require existence of constants $C, \alpha_{\mathcal{X}}$ and $\alpha_{\mathcal{Y}}$, such that all eigenvalues of $C_{\mathcal{X}}$ and $C_{\mathcal{Y}}$ are upper and lower bounded as,

$$(1/C)j^{-\alpha_{\mathcal{X}}} \leq \lambda_j(C_{\mathcal{X}}) \leq Cj^{-\alpha_{\mathcal{X}}} \tag{3}$$
$$(1/C)j^{-\alpha_{\mathcal{Y}}} \leq \lambda_j(C_{\mathcal{Y}}) \leq Cj^{-\alpha_{\mathcal{Y}}}$$

The statistical rate then, in the sense of convergence of parameters, $\|C_{\mathcal{X}}^{1/2}(u_1 - \hat{u}_1)\|_{\mathcal{H}_{\mathcal{X}}}^2 = O_p(\max(n^{-\alpha_{\mathcal{X}}/(\alpha_{\mathcal{X}}+1)}, n^{-\alpha_{\mathcal{Y}}/(\alpha_{\mathcal{Y}}+1)}))$. As remarked before, these asymptotic guarantees do not quantify the number of samples required in terms of the confidence parameter $\delta$, and are even weaker than convergence in mean. Moreover, there is a *large* regime where their additional assumptions do not hold, but our results still apply. In particular, suppose that eigenvalues decay exponentially, i.e., $\lambda_j(C_{\mathcal{X}}) = e^{-j}$. Since exponential decays faster than any inverse polynomial, so it goes below the inverse polynomial for large enough $j$ (and we are in infinite dimensions) - this violates the condition of Fan and Lian (2016). In general, any sub-polynomial or super-polynomial eigenvalue value behavior of $C_{\mathcal{X}}$ or $C_{\mathcal{Y}}$ violates their condition; but our results are agnostic to it. Finally, their result doesn't provide any insights on how the regularization parameter be set to control and bias and variance, which is important from a practical perspective.

**Regularization parameter.** Fukumizu et al. (2007) suggest setting the regularization parameter as $\omega(n^{-1/3})$ to ensure statistical consistency of the ERM. In contrast, when we set $\lambda_{\mathcal{X}} = \lambda_{\mathcal{Y}} = \lambda$, our results showcase an improvement of the same estimator to $\omega(n^{-1/2})$, together with high-confidence guarantees. This follows because bias is in $O(\sqrt{\lambda})$ and variance is of the order $O(\lambda^{-1}n^{-1/2})$, so both decrease with $n$ when $\lambda = \omega(n^{-1/2})$.

**Finite dimensional Hilbert spaces.** As a special case, our result can be applied to give guarantees for (unregularized) ERM wherein we assume that the smallest eigenvalues of auto-covariance operators $C_{\mathcal{X}}$ and $C_{\mathcal{Y}}$ are lower bounded away from 0 by $\lambda_{\mathcal{X}}$ and $\lambda_{\mathcal{Y}}$, respectively. This subsumes standard CCA problem as well as kernel CCA where the RKHS corresponding to the kernel map is finite dimensional. We obtain rates ranging from $n^{-1/2}$ to $n^{-1}$ depending on the spectrum decay of $\mathfrak{C}'$.

**Corollary 3** (Finite dimensional kernel CCA)**.** *Along with the notations and assumptions of Corollary 2, suppose that RKHS $\mathcal{H}_{\mathcal{X}}$ and $\mathcal{H}_{\mathcal{Y}}$ are finite dimensional with the eigenvalues of $C_{\mathcal{X}}$ and $C_{\mathcal{Y}}$ lower bounded by $\lambda_{\mathcal{X}} > 0$ and $\lambda_{\mathcal{Y}} > 0$ respectively. Then, ERM on $n$ training data outputs $U_n V_n^*$ which satisfies the following with probability at least $1 - \delta$,*

$$\left\langle U_C V_C^* - U_{n,\lambda} V_{n,\lambda}^*, C \right\rangle_{L(\mathcal{H}_{\mathcal{Y}}, \mathcal{H}_{\mathcal{X}})} \leq O\left( \frac{\log(1/\delta)}{gap_k(C)\lambda^2 n} + \inf_{h \geq 0} \left\{ \frac{h}{gap_k(C)\lambda^2 n} + \frac{1}{\lambda\sqrt{n}}\sqrt{\sum_{j>h} \lambda_j(\mathfrak{C}')} \right\} \right)$$

*where $gap_k(C) = \lambda_k(C) - \lambda_{k+1}(C)$ is the eigengap of $C$ at $k$.*

**Linear CCA.** We compare our results with that of Gao et al. (2017) for the special case of the linear CCA. For $k = 1$, Gao et al. (2017) showed that ERM achieves $\epsilon$ sub-optimality with $O(1/gap^2\lambda^2\epsilon)$ sample complexity, where gap $= \lambda_1(C) - \lambda_2(C)$, ignoring log factors. In comparison, our sample complexity is $O(1/gap\lambda^2\epsilon)$ in the optimistic case and $O(1/\lambda^2\epsilon\min\{gap, \epsilon\})$ in the worst case; therefore, our optimistic case is *always* better than Gao et al. (2017), whereas the general/worst-case rate is better whenever gap $= \omega(\sqrt{\epsilon})$.

See Table 1 for a summary of our results and comparison with prior works.

---

[1] $O_p$ and $o_P$ are standard asymptotic notation for rates of convergence in probability - for a pair of random sequences, $f_n$ and $g_n$, we write $f_n = O_p(g_n)$ if $f_n/|g_n|$ is bounded in probability, and $f_n = o_p(g_n)$ if $f_n/|g_n|$ converges to 0, in probability.

## 5 Proof Sketch

In this section, we sketch the main proof ideas underlying our analysis; full details are deferred to Appendix B. Recall that the optimal solution to the population objective are given by operators $U_C$ and $V_C$ that correspond to the first $k$ eigenfunctions of $C^*C$ and $CC^*$, where $C = C_{\mathcal{X}}^{-1/2} C_{\mathcal{X}\mathcal{Y}} C_{\mathcal{Y}}^{-1/2}$. For the sake of analysis, we introduce regularized whitened population cross-covariance operator $C_\lambda = (C_{\mathcal{X}} + \lambda_{\mathcal{X}})^{-1/2} C_{\mathcal{X}\mathcal{Y}} (C_{\mathcal{Y}} + \lambda_{\mathcal{Y}})^{-1/2}$, whose top $k$ left and right singular functions are denoted by $U_{C,\lambda}$ and $V_{C,\lambda}$. The empirical counterparts are denoted by $C^n = (C_{\mathcal{X}}^n)^{-1/2} C_{\mathcal{X}\mathcal{Y}}^n (C_{\mathcal{Y}}^n)^{-1/2}$, $U_n$ and $V_n$, and the regularized empirical counterparts by $C^{n,\lambda} = (C_{\mathcal{X}}^{n,\lambda_{\mathcal{X}}})^{-1/2} C_{\mathcal{X}\mathcal{Y}}^n (C_{\mathcal{Y}})^{-1/2}$, $U_{n,\lambda}$ and $V_{n,\lambda}$. We begin with the following decomposition of the excess error into approximation error, and estimation error,

$$
\begin{aligned}
\mathcal{E}(U_{n,\lambda}, V_{n,\lambda}) = \left\langle U_C V_C^* - U_{n,\lambda} V_{n,\lambda}^*, C \right\rangle_{L(\mathcal{H}_{\mathcal{Y}}, \mathcal{H}_{\mathcal{X}})} &= \underbrace{\left\langle U_C V_C^* - U_{C,\lambda} V_{C,\lambda}^*, C \right\rangle_{L(\mathcal{H}_{\mathcal{Y}}, \mathcal{H}_{\mathcal{X}})}}_{\textbf{Approximation error}} \\
+ \underbrace{\left\langle U_{C,\lambda} V_{C,\lambda}^* - U_{n,\lambda} V_{n,\lambda}^*, C \right\rangle_{L(\mathcal{H}_{\mathcal{Y}}, \mathcal{H}_{\mathcal{X}})}}_{\textbf{Estimation error}} &= \underbrace{\left\langle U_C V_C^* - U_{C,\lambda} V_{C,\lambda}^*, C \right\rangle_{L(\mathcal{H}_{\mathcal{Y}}, \mathcal{H}_{\mathcal{X}})}}_{\textbf{Approximation error}} \\
+ \underbrace{\left\langle U_{C,\lambda} V_{C,\lambda}^* - U_{n,\lambda} V_{n,\lambda}^*, C - C_\lambda \right\rangle_{L(\mathcal{H}_{\mathcal{Y}}, \mathcal{H}_{\mathcal{X}})}}_{\textbf{Estimation error 2}} &+ \underbrace{\left\langle U_{C,\lambda} V_{C,\lambda}^* - U_{n,\lambda} V_{n,\lambda}^*, C_\lambda \right\rangle_{L(\mathcal{H}_{\mathcal{Y}}, \mathcal{H}_{\mathcal{X}})}}_{\textbf{Error 3}}
\end{aligned}
$$

We can write the sum of the first two terms, i.e., the approximation error and the estimation error 2 as

$$
\underbrace{\left\langle U_C V_C^*, C \right\rangle_{L(\mathcal{H}_{\mathcal{Y}}, \mathcal{H}_{\mathcal{X}})} - \left\langle U_{C,\lambda} V_{C,\lambda}^*, C_\lambda \right\rangle_{L(\mathcal{H}_{\mathcal{Y}}, \mathcal{H}_{\mathcal{X}})}}_{\textbf{Error 1}} + \underbrace{\left\langle U_{n,\lambda} V_{n,\lambda}^*, C_\lambda - C \right\rangle_{L(\mathcal{H}_{\mathcal{Y}}, \mathcal{H}_{\mathcal{X}})}}_{\textbf{Error 2}}
$$

We bound each of the three error terms separately. The bound on Error 1 and Error 2 (see Appendix B) holds due to the following lemma which bounds the distance between the whitened covariance operator and its regularized counterpart.

**Lemma 4.** *Let $C = C_{\mathcal{X}}^{-1/2} C_{\mathcal{X}\mathcal{Y}} C_{\mathcal{Y}}^{-1/2}$ and $C_\lambda = (C_{\mathcal{X}} + \lambda_{\mathcal{X}} I_{\mathcal{X}})^{-1/2} C_{\mathcal{X}\mathcal{Y}} (C_{\mathcal{Y}} + \lambda_{\mathcal{Y}} I_{\mathcal{Y}})^{-1/2}$ be its regularized counterpart. Suppose that $C$ is compact, then, there difference is bounded as, $\|C - C_\lambda\| \leq 4(\sqrt{\lambda_{\mathcal{X}}} + \sqrt{\lambda_{\mathcal{Y}}})$.*

It remains to bound Error 3; we prove the following result which follows using the local Rademacher complexity analysis of Bartlett et al. (2002).

**Lemma 5** (Error 3). *With probability at least $1 - \delta$, the Error 3 is bounded as,*

$$
\left\langle U_{C,\lambda} V_{C,\lambda}^* - U_{n,\lambda} V_{n,\lambda}^*, C_\lambda \right\rangle_{L(\mathcal{H}_{\mathcal{Y}}, \mathcal{H}_{\mathcal{X}})} \leq \inf_{h \geq 0} \left\{ \frac{12\alpha_\rho h}{\lambda_{\mathcal{X}} \lambda_{\mathcal{Y}} n} + \frac{24\alpha_\rho \sqrt{h}}{\lambda_{\mathcal{X}} \lambda_{\mathcal{Y}} n} + \frac{24}{\sqrt{n}} \sqrt{\frac{k}{\lambda_{\mathcal{X}} \lambda_{\mathcal{Y}}} \sum_{j > h} \lambda_j(\mathfrak{C}')} \right\}
$$
$$
+ \frac{12\alpha_\rho}{\lambda_{\mathcal{X}} \lambda_{\mathcal{Y}} n} + \frac{22\beta \sqrt{k} \log(1/\delta)}{\sqrt{\lambda_{\mathcal{X}} \lambda_{\mathcal{Y}} n}} + \frac{10\alpha_\rho \log(1/\delta)}{\lambda_{\mathcal{X}} \lambda_{\mathcal{Y}} n}
$$

*where $\alpha_\rho = \mathbb{E}_{x,y,x',y'} \left[ \langle x, x' \rangle_{\mathcal{H}_{\mathcal{X}}}^2 \langle y, y' \rangle_{\mathcal{H}_{\mathcal{Y}}}^2 \right] / (\lambda_k(C) - \lambda_{k+1}(C))$.*

We sketch a proof of the key result in the next section.

### 5.1 Proof Sketch of Lemma 5

Our main tool is local Rademacher complexity analysis (Bartlett et al., 2002) of the problem, following its application in Kernel PCA (Blanchard et al., 2007). The exact statement of local Rademacher complexity result of (Bartlett et al., 2002) is stated as Theorem 10 in Appendix B. We now discuss how we apply it to our end. We first define the excess risk function class.

$$
\mathcal{F} = \left\{ f_{U,V} : (x, y) \to \left\langle \overline{U}_{C,\lambda} \overline{V}_{C,\lambda}^* - UV^*, C_{x,y} \right\rangle_{L(\mathcal{H}_{\mathcal{Y}}, \mathcal{H}_{\mathcal{X}})} \mid U^* C_{\mathcal{X}}^{\lambda_{\mathcal{X}}} U = I_k, V^* C_{\mathcal{Y}}^{\lambda_{\mathcal{Y}}} V = I_k \right\},
$$

where $\overline{\mathrm{U}}_{C,\lambda} = \left(\mathrm{C}_{\mathcal{X}}^{\lambda_{\mathcal{X}}}\right)^{1/2} \mathrm{U}_{C,\lambda}, \overline{\mathrm{V}}_{C,\lambda} = \left(\mathrm{C}_{\mathcal{Y}}^{\lambda_{\mathcal{Y}}}\right)^{1/2} \mathrm{V}_{C,\lambda}$ and $\mathrm{C}_{\mathrm{x,y}} = \mathrm{x} \otimes_{L(\mathcal{H}_{\mathcal{Y}}, \mathcal{H}_{\mathcal{X}})} \mathrm{y}$.

We remark that we cannot directly apply the local Rademacher complexity technique to this class as Theorem 10 requires that the range of functions be contained in $[-1, 1]$. We therefore look at $\mathcal{G} = \tau\mathcal{F}$, where $\tau = \sqrt{\lambda_{\mathcal{X}}\lambda_{\mathcal{Y}}}/(2\beta\sqrt{k})$ and show that it satisfies this requirement.

**Lemma 6.** *For any $f \in \mathcal{G} = \tau\mathcal{F}$ the range of $f$ is contained in $[-1, 1]$, where $\tau = \frac{\sqrt{\lambda_{\mathcal{X}}\lambda_{\mathcal{Y}}}}{2\beta\sqrt{k}}$.*

We also need to show that this function class $\mathcal{G}$ satisfies the "local" assumption of Theorem 10, i.e., the second moment is at most a multiple of mean.

**Lemma 7.** *For any $f \in \mathcal{G}$, $\mathbb{E}\left[f^2\right] \leq \mu\,\mathbb{E}\left[f\right]$ where $\mu = 2\alpha_\rho\tau/(\lambda_{\mathcal{X}}\lambda_{\mathcal{Y}})$, and $\alpha_\rho = \mathbb{E}_{\mathrm{x,y,x',y'}}\left[\langle\mathrm{x,x'}\rangle_{\mathcal{H}_{\mathcal{X}}}^2 \langle\mathrm{y,y'}\rangle_{\mathcal{H}_{\mathcal{Y}}}^2\right]/(\lambda_k(\mathrm{C}) - \lambda_{k+1}(\mathrm{C}))$.*

Next, we bound the Rademacher complexity of the star shaped hull of the function class conditioned on bounded second moment. We simplify this by considering a bigger class which contains $\mathrm{star}(\mathcal{G})$. Formally, we show that the family $\left\{g \in \mathrm{star}(\mathcal{G}) \,\middle|\, \mathbb{E}\left[g^2\right] \leq r\right\}$ is contained in

$$\mathcal{S}_r := \tau\left\{(x,y) \to \langle\Gamma, \mathrm{C}_{x,y}\rangle_{L(\mathcal{H}_{\mathcal{Y}}, \mathcal{H}_{\mathcal{X}})} \,\middle|\, \Gamma \in L(\mathcal{H}_{\mathcal{Y}}, \mathcal{H}_{\mathcal{X}}), \|\Gamma\|_{L(\mathcal{H}_{\mathcal{Y}}, \mathcal{H}_{\mathcal{X}})}^2 \leq \frac{4k}{\lambda_{\mathcal{X}}\lambda_{\mathcal{Y}}}, \langle\Gamma, \mathfrak{C}\Gamma\rangle_{L(\mathcal{H}_{\mathcal{Y}}, \mathcal{H}_{\mathcal{X}})} \leq \frac{r}{\tau^2}\right\},$$

where $\mathfrak{C}' : L(\mathcal{H}_{\mathcal{Y}}, \mathcal{H}_{\mathcal{X}}) \to L(\mathcal{H}_{\mathcal{Y}}, \mathcal{H}_{\mathcal{X}})$ is a fourth moment operator, defined as $\mathfrak{C}' := \mathbb{E}_{\mathrm{x,y}}\left[\left(\mathrm{x} \otimes_{L(\mathcal{H}_{\mathcal{Y}}, \mathcal{H}_{\mathcal{X}})} \mathrm{y}\right) \otimes_{L(L(\mathcal{H}_{\mathcal{Y}}, \mathcal{H}_{\mathcal{X}}))} \left(\mathrm{x} \otimes_{L(\mathcal{H}_{\mathcal{Y}}, \mathcal{H}_{\mathcal{X}})} \mathrm{y}\right)\right]$. A crucial technical result follows.

**Lemma 8.** *The Rademacher complexity of $\mathcal{S}_r$ is bounded as follows,*

$$\mathfrak{R}_n(\mathcal{S}_r) \leq \sqrt{\frac{r}{n}} + \frac{1}{\sqrt{n}} \inf_{h \geq 0} \left\{\sqrt{r}h + 2\tau\sqrt{\frac{k}{\lambda_{\mathcal{X}}\lambda_{\mathcal{Y}}}\sum_{j>h}\lambda_j(\mathfrak{C}')}\right\}$$

*where $\mathfrak{C}' = \mathfrak{C} - \mathrm{C}_{\mathcal{X}\mathcal{Y}} \otimes_{L(L(\mathcal{H}_{\mathcal{Y}}, \mathcal{H}_{\mathcal{X}}))} \mathrm{C}_{\mathcal{X}\mathcal{Y}}$.*

A final requirement to apply Theorem 10 is that $\psi(r) := \mu\,\mathfrak{R}_n(\mathcal{S}_r)$ is a sub-root function in $r$. This simply follows from the above lemma and the fact that the pointwise infimum of sub-root functions is a sub-root function. Furthermore, the existence of an upper bound on the fixed point is again guaranteed because it is a sub-root function. In particular, we show the following.

**Lemma 9.** *The fixed point of $\psi(r)$, denoted as $r^*$, is bounded as,*

$$r^* \leq \tau^2\left(\inf_{h \geq 0}\left\{\frac{\xi^2 h}{n} + \frac{2\xi^2\sqrt{h}}{n} + \frac{4\xi}{\sqrt{m}}\sqrt{\frac{k}{\lambda_{\mathcal{X}}\lambda_{\mathcal{Y}}}\sum_{j>h}\lambda_j(\mathfrak{C}')}\right\} + \frac{\xi^2}{n}\right)$$

*where $\xi = \frac{2\alpha_\rho}{\lambda_{\mathcal{X}}\lambda_{\mathcal{Y}}}$, and $\alpha_\rho = \mathbb{E}_{(\mathrm{x,y})\sim\rho,(\mathrm{x',y'})\sim\rho}[\langle\mathrm{x,x'}\rangle_{\mathcal{H}_{\mathcal{X}}}^2 \langle\mathrm{y,y'}\rangle_{\mathcal{H}_{\mathcal{Y}}}^2]/(\lambda_k(\mathrm{C}) - \lambda_{k+1}(\mathrm{C}))$.*

To finish the proof, we set $\mathrm{U} = (\mathrm{C}_{\mathcal{X}}^{\lambda_{\mathcal{X}}})^{-1/2}\mathrm{U}_{n,\lambda}$ and $\mathrm{V} = (\mathrm{C}_{\mathcal{Y}}^{\lambda_{\mathcal{Y}}})^{-1/2}\mathrm{V}_{n,\lambda}$, where $\mathrm{U}_{n,\lambda}$ and $\mathrm{V}_{n,\lambda}$ are the solutions to the regularized ERM problem. Note that $\mathbb{E}_n[f_{\mathrm{U,V}}] \leq 0$ where $\mathbb{E}_n$ is expectation with respect to the empirical measure. Applying Theorem 10, stated in Appendix B, and letting $K \to 1$ yields Theorem 1.

# 6 Conclusion

In this work, we established finite sample generalization bounds on regularized ERM for functional and kernel CCA. The focus here was on understanding the statistical complexity of regularized ERM for kernel CCA. A promising future direction is to study the problem from an algorithmic and computational perspective, in particular when using approximate feature maps based on random Fourier features (Rahimi and Recht, 2007), Nyström's method (Drineas and Mahoney, 2005) or with randomized sketching (Yang et al., 2015).

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

# Supplementary Material

## A    Preliminaries

### A.1    Random variables in Hilbert spaces

We start with briefly discussing probability theory in Hilbert spaces. Let $(\mathcal{H}_\mathcal{X}, \mathcal{F}_\mathcal{X}, \rho_\mathcal{X})$ and $(\mathcal{H}_\mathcal{X}, \mathcal{F}_\mathcal{X}, \rho_\mathcal{X})$ be measurable Hilbert spaces. A random variable x in $\mathcal{H}_\mathcal{X}$ is well-defined if and only if every continuous form $\langle f, \mathrm{x} \rangle_{\mathcal{H}_\mathcal{X}}$ is measurable for all $f \in \mathcal{H}_\mathcal{X}$. Its expectation is $\mu_\mathcal{X} \in \mathcal{H}_\mathcal{X}$ if $\langle \mu_\mathcal{X}, f \rangle_{\mathcal{H}_\mathcal{X}} = \mathbb{E}_{\rho_\mathcal{X}} \left[ \langle \mathrm{x}, f \rangle_{\mathcal{H}_\mathcal{X}} \right] \ \forall \ f \in \mathcal{H}_\mathcal{X}$. The existence and uniqueness of $\mu_\mathcal{X}$ is guaranteed if $\|\mathrm{x}\|_{\mathcal{H}_\mathcal{X}} < \infty$. We similarly consider the space $(\mathcal{H}_\mathcal{Y}, \mathcal{F}_\mathcal{Y}, \rho_\mathcal{Y})$ with random variable y. Throughout the paper, we assume that the random variable x has mean $\mathbf{0}_\mathcal{X} \in \mathcal{H}_\mathcal{X}$, formally defined as $\mathbb{E}_\mathrm{x} \left[ \langle \mathrm{x}, f \rangle_{\mathcal{H}_\mathcal{X}} \right] = \langle \mathbf{0}_\mathcal{X}, f \rangle_{\mathcal{H}_\mathcal{X}} = 0 \ \forall \ f \in \mathcal{H}_\mathcal{X}$, to simplify the presentation. Similarly, the mean of y is assumed to be $\mathbf{0}_\mathcal{Y}$.

**Definition 1** (Auto-covariance operator). *Let* x *be a random variable in a Hilbert space* $\mathcal{H}_\mathcal{X}$ *such that* $\|\mathrm{x}\|_{\mathcal{H}_\mathcal{X}} < \infty$, *with distribution* $\rho_\mathcal{X}$ *Then, the auto-covariance operator of* x, *denoted as* $\mathrm{C}_\mathcal{X} : \mathcal{H}_\mathcal{X} \to \mathcal{H}_\mathcal{X}$ *is defined as,* $\forall \ f, g \in \mathcal{H}_\mathcal{X}$,

$$\langle f, \mathrm{C}_\mathcal{X} g \rangle = \int_\mathcal{X} \langle f, \mathrm{x} \rangle_{\mathcal{H}_\mathcal{X}} \langle g, \mathrm{x} \rangle_{\mathcal{H}_\mathcal{X}} \, d\rho_\mathcal{X}(\mathrm{x}).$$

*Equivalently,* $\mathrm{C}_\mathcal{X} := \mathbb{E}_{\rho_\mathcal{X}} \left[ \mathrm{x} \otimes_{L(\mathcal{H}_\mathcal{X})} \mathrm{x} \right]$. *Furthermore,* $\mathrm{C}_\mathcal{X}$ *is self-adjoint, positive and trace class.*

We similarly use $\mathrm{C}_\mathcal{Y}$ to denote the auto-covariance operator of random variable y in $\mathcal{H}_\mathcal{Y}$.

Let $\mathcal{H}_\mathcal{X} \times \mathcal{H}_\mathcal{Y}$ be the product space of $\mathcal{H}_\mathcal{X}$ and $\mathcal{H}_\mathcal{Y}$, and let $\mathcal{F}_\mathcal{X} \times \mathcal{F}_\mathcal{Y}$ be the $\sigma$-field generated by the product of elements of $\mathcal{F}_\mathcal{X}$ and $\mathcal{F}_\mathcal{Y}$. Let $\rho$ be the joint probability measure on $(\mathcal{H}_\mathcal{X} \times \mathcal{H}_\mathcal{Y}, \mathcal{F}_\mathcal{X} \times \mathcal{F}_\mathcal{Y})$ with its marginal/projection on $(\mathcal{H}_\mathcal{X}, \mathcal{F}_\mathcal{X})$ and $(\mathcal{H}_\mathcal{Y}, \mathcal{F}_\mathcal{Y})$ being $\rho_\mathcal{X}$ and $\rho_\mathcal{Y}$ respectively. We now define a cross-covariance operator (Baker, 1973).

**Definition 2** (Cross-covariance operator). *Let* $(\mathrm{x}, \mathrm{y})$ *be paired random variables in* $\mathcal{H}_\mathcal{X} \times \mathcal{H}_\mathcal{Y}$ *distributed as* $\rho$, *such that* $\|\mathrm{x}\|_{\mathcal{H}_\mathcal{X}} < \infty$ *and* $\|\mathrm{y}\|_{\mathcal{H}_\mathcal{Y}} < \infty$. *The cross-covariance operator* $\mathrm{C}_{\mathcal{X}\mathcal{Y}} : \mathcal{H}_\mathcal{Y} \to \mathcal{H}_\mathcal{X}$ *is defined as*

$$\langle f, \mathrm{C}_{\mathcal{X}\mathcal{Y}} g \rangle_{\mathcal{H}_\mathcal{Y}} = \mathbb{E}_\rho \left[ \langle f, \mathrm{x} \rangle_{\mathcal{H}_\mathcal{X}} \langle g, \mathrm{y} \rangle_{\mathcal{H}_\mathcal{Y}} \right].$$

*Equivalently,* $\mathrm{C}_{\mathcal{X}\mathcal{Y}} := \mathbb{E}_\rho \left[ \mathrm{x} \otimes_{L(\mathcal{H}_\mathcal{Y}, \mathcal{H}_\mathcal{X})} \mathrm{y} \right]$. *The adjoint of* $\mathrm{C}_{\mathcal{X}\mathcal{Y}}$, *denoted as* $\mathrm{C}_{\mathcal{X}\mathcal{Y}}^* = \mathrm{C}_{\mathcal{Y}\mathcal{X}}$, *which is defined similarly.*

The existence and uniqueness of bounded auto-covariance and cross-covariance operators is guaranteed by Riesz's representation theorem (Reed and Simon, 1972).

### A.2    Reproducing Kernel Hilbert Spaces

Let $\mathcal{X} \subseteq \mathbb{R}^{d_\mathcal{X}}$ and $\mathcal{Y} \subseteq \mathbb{R}^{d_\mathcal{Y}}$ be finite dimensional data domains corresponding to the two views. Let $\rho$ be the joint distribution over $\mathcal{X} \times \mathcal{Y}$ and $\rho_\mathcal{X}$ and $\rho_\mathcal{Y}$ denote the marginals. We start with a definition of a reproducing kernel. Our discussion is in the context of $k_\mathcal{X}$ but it analogously holds for $k_\mathcal{Y}$ as well.

**Definition 3** (Reproducing kernel (Sejdinovic and Gretton, 2012)). *Let* $\mathcal{H}_\mathcal{X}$ *be a Hilbert space of real valued functions over* $\mathcal{X}$. *A function* $k_\mathcal{X} : \mathcal{X} \times \mathcal{X} \to \mathbb{R}$ *is called a reproducing kernel of* $\mathcal{H}_\mathcal{X}$, *if the following hold,*

- $\forall \ \mathrm{x} \in \mathcal{X}, \ k_\mathcal{X}(\mathrm{x}, \cdot) \in \mathcal{H}_\mathcal{X}$
- $\forall \ \mathrm{x} \in \mathcal{X}, \ \forall \ f \in \mathcal{H}_\mathcal{X}, \langle f, k_\mathcal{X}(\mathrm{x}, \cdot) \rangle_{\mathcal{H}_\mathcal{X}} = f(\mathrm{x})$ *(Reproducing property)*

The function $k_\mathcal{X}(\mathrm{x}, \cdot)$ is the *representer* of the evaluation functional at x. The inner product between two elements can therefore be expressed as $\langle k_\mathcal{X}(\mathrm{x}, \cdot), k_\mathcal{X}(\mathrm{x}', \cdot) \rangle_{\mathcal{H}_\mathcal{X}} = k_\mathcal{X}(\mathrm{x}, \mathrm{x}')$. Moreover, reproducing kernels are always positive definite.

A Reproducing Kernel Hilbert Space, conventionally abbreviated as RKHS, is defined as the completion of the linear span of $\{k_{\mathcal{X}}(\mathrm{x},\cdot),\ \mathrm{x} \in \mathcal{X}\}$ with respect to the inner product $\langle k_{\mathcal{X}}(\mathrm{x},\cdot), k_{\mathcal{X}}(\mathrm{x}',\cdot)\rangle_{\mathcal{H}_{\mathcal{X}}} = k_{\mathcal{X}}(\mathrm{x},\mathrm{x}')$. The existence and uniqueness of a reproducing kernel for an RKHS is guaranteed by Riesz-representation theorem. We point the interested reader to Sejdinovic and Gretton (2012) for more discussion on RKHS.

We now briefly discuss auto-covariance and cross-covariance operators in the context of RKHS. As before, we assume that $\sup_{\mathrm{x}} \|k_{\mathcal{X}}(\mathrm{x},\cdot)\|_{\mathcal{H}_{\mathcal{X}}} < \infty$ and $\sup_{\mathrm{y}} \|k_{\mathcal{Y}}(\mathrm{y},\cdot)\|_{\mathcal{H}_{\mathcal{Y}}} < \infty$. The auto covariance operator over $\mathcal{H}_{\mathcal{X}}$, $\mathrm{C}_{\mathcal{X}} : \mathcal{H}_{\mathcal{X}} \to \mathcal{H}_{\mathcal{X}}$ is defined as, $\langle f_1, \mathrm{C}_{\mathcal{X}} f_2\rangle_{\mathcal{H}_{\mathcal{X}}} = \mathbb{E}_{\rho_{\mathcal{X}}}[f_1(\mathrm{x})\ f_2(\mathrm{x})]\ \forall f_1, f_2 \in \mathcal{H}_{\mathcal{X}}$ or equivalently, $\mathrm{C}_{\mathcal{X}} := \mathbb{E}_{\rho_{\mathcal{X}}}\left[k_{\mathcal{X}}(\mathrm{x},\cdot) \otimes_{L(\mathcal{H}_{\mathcal{X}})} k_{\mathcal{X}}(\mathrm{x},\cdot)\right]$. Similarly, $\mathrm{C}_{\mathcal{Y}} : \mathcal{H}_{\mathcal{Y}} \to \mathcal{H}_{\mathcal{Y}}$ is defined as, $\langle g_1, \mathrm{C}_{\mathcal{Y}} g_2\rangle_{\mathcal{H}_{\mathcal{X}}} = \mathbb{E}_{\rho_{\mathcal{Y}}}[g_1(\mathrm{y})\ g_2(\mathrm{y})]\ \forall g_1, g_2 \in \mathcal{H}_{\mathcal{X}}$ or equivalently, $\mathrm{C}_{\mathcal{Y}} := \mathbb{E}_{\rho_{\mathcal{Y}}}\left[k_{\mathcal{Y}}(\mathrm{y},\cdot) \otimes_{L(\mathcal{H}_{\mathcal{Y}})} k_{\mathcal{Y}}(\mathrm{y},\cdot)\right]$ Furthermore, the cross-covariance operator $\mathrm{C}_{\mathcal{X}\mathcal{Y}} : \mathcal{H}_{\mathcal{Y}} \to \mathcal{H}_{\mathcal{X}}$ is defined as, $\langle f, \mathrm{C}_{\mathcal{X}\mathcal{Y}} g\rangle_{\mathcal{H}_{\mathcal{X}}} = \mathbb{E}_{\rho}[f(\mathrm{x})\ g(\mathrm{y})]$ or equivalently, $\mathrm{C}_{\mathcal{X}\mathcal{Y}} := \mathbb{E}_{\rho}\left[k_{\mathcal{X}}(\mathrm{x},\cdot) \otimes_{L(\mathcal{H}_{\mathcal{Y}},\mathcal{H}_{\mathcal{X}})} k_{\mathcal{Y}}(\mathrm{y},\cdot)\right]$. These are easy to see from definitions 1 and 2 and applying reproducing property from definition 3.

**Kernel CCA notation.** In the context of kernel CCA, we setup the following notation. Let $\mathcal{X} \subseteq \mathbb{R}^{d_{\mathcal{X}}}$ and $\mathcal{Y} \subseteq \mathbb{R}^{d_{\mathcal{Y}}}$ be finite dimensional data domains corresponding to the two views. Let $\rho$ be the joint distribution over $\mathcal{X} \times \mathcal{Y}$ and $\rho_{\mathcal{X}}$ and $\rho_{\mathcal{Y}}$ denote the marginals. Let $\{(\mathrm{x}_i, \mathrm{y}_i)\}_{i=1 \text{ to } n}$ be $n$ data points drawn from $\rho$. Let $k_{\mathcal{X}}$ and $k_{\mathcal{Y}}$ be two kernel functions and $\mathcal{H}_{\mathcal{X}}$ and $\mathcal{H}_{\mathcal{Y}}$ be the RKHS associated with $(\mathcal{X}, k_{\mathcal{X}})$ and $(\mathcal{Y}, k_{\mathcal{Y}})$ respectively.

### A.3 Local Rademacher Complexity Technique

Rademacher complexity is a data-dependent notion of complexity which captures the richness of a class of real valued functions with respect to the data distribution. It is a fundamental concept in statistical learning theory and empirical process theory, and enables uniform convergence guarantees for the learning problem. We define it formally below.

**Definition 4** (Rademacher Complexity). *Let $\mathcal{F}$ be a set of functions over $\mathcal{X}$ and let $\rho_{\mathcal{X}}$ be a probability distribution over $\mathcal{X}$. For a positive integer $n$, let $\mathrm{x}_1, \mathrm{x}_2, \ldots, \mathrm{x}_n$ be i.i.d. samples drawn from $\rho_{\mathcal{X}}$ and let $\sigma_1, \sigma_2, \ldots, \sigma_n$ be i.id. samples drawn from Rademacher distribution. The Rademacher complexity of $\mathcal{F}$, denoted by $\mathfrak{R}_n(\mathcal{F})$, is defined as,*

$$\mathfrak{R}_n(\mathcal{F}) = \mathbb{E}_{\sigma, \mathrm{x}}\left[\sup_{f \in \mathcal{F}} \frac{1}{n}\sum_{i=1}^{n} \sigma_i f(\mathrm{x}_i)\right].$$

Local Rademacher complexity technique is a concentration of measure tool, introduced in Bartlett et al. (2002), which aims to provide a *finer* analysis of the problem. It is motivated from the claim that the hypothesis selected by a learning algorithm usually has a low empirical error, and hence looking at the Rademacher complexity of the whole class is wasteful. Bartlett et al. (2002) use the variance of the empirical process to constrain the hypothesis class. In particular, if the variance is upper bounded by a constant multiple of the mean, then the empirical process is well-behaved and admits faster rates. This is formalized in Theorem 10 (in Section B) which is restated from Bartlett et al. (2002).

Local Rademacher complexity technique, therefore, essentially looks at the Rademacher complexity of a *smaller* subset of functions from the original hypothesis class. Formally, we have $\mathfrak{R}_n(\mathcal{F}, r) = \mathfrak{R}_n(\{f \in \mathrm{star}(\mathcal{F}) \,|\, \mathbb{E}\left[f^2\right] \le r\})$ where $\mathrm{star}(\mathcal{F}) = \{\alpha f \,|\, f \in \mathcal{F},\ \alpha \in [0,1]\}$ is the star-hull of $\mathcal{F}$. This technique has been used to derive *sharper* generalization bounds for various kernel learning problems (Blanchard et al., 2007; Ullah et al., 2018; Cortes et al., 2013; Gilbert et al., 2018). Informally, the rate obtained is usually controlled by the tail decay of the spectrum of a *higher order moment*. We discuss this in more detail in subsequent sections.

## B  Proof of the main theorem

We first restate the local Rademacher complexity result from Bartlett et al. (2002), which is a key tool in our analysis.

**Theorem 10.** *(Bartlett et al., 2002) Let $\mathcal{X}$ be a measurable space and let $\mathcal{P}$ be a probability distribution on $\mathcal{X}$. Let $x_1, x_2 \ldots x_n$ be i.i.d. samples drawn from $\mathcal{P}$, let $\mathcal{P}_n$ denote its empirical measure, and let $\mathcal{P}f := \mathbb{E}_{x \sim \mathcal{P}}[f(x)]$ and $\mathcal{P}_n f := \frac{1}{n} \sum_{i=1}^n f(x_i)$ for a measurable function $f$. Let $\mathcal{F}$ be a class of functions on $\mathcal{X}$ ranging from $[-1, 1]$ and assume that there exists some constant $B$ such that for every $f \in \mathcal{F}, \mathcal{P}^2 f \leq B\mathcal{P}f$. Let $\psi$ be a sub-root function and let $r^*$ be the fixed point of $\psi$. Suppose that $\psi$ satisfies*

$$\psi(r) \geq B \, \mathfrak{R}_n \{ f \in star(\mathcal{F}) | \mathcal{P}f^2 \leq r \}$$

*where $star(\mathcal{F}) = \{ \lambda f \mid f \in \mathcal{F}, \lambda \in [0,1] \}$ is the star shaped hull of $\mathcal{F}$ and $\mathfrak{R}_n(\mathcal{F}) = \mathbb{E}_{\sigma,x} \left[ \sup_{f \in \mathcal{F}} \frac{1}{n} \sum_{i=1}^n \sigma_i f(x_i) \right]$ is the Rademacher complexity of $\mathcal{F}$ given $n$ data points from $\mathcal{P}$, then for every $K > 0$ and $\delta > 0$, with probability at least $1 - e^{-\delta}$*

$$\forall \; f \in \mathcal{F}, \mathcal{P}f \leq \frac{K}{K-1}\mathcal{P}_n f + \frac{6K}{B}r^* + \frac{\delta(11 + 5BK)}{n} \tag{4}$$

*Also, with probability at least $1 - e^{-\delta}$*

$$\forall f \in \mathcal{F}, \mathcal{P}_n f \leq \frac{K}{K+1}\mathcal{P}f + \frac{6K}{B}r^* + \frac{\delta(11 + 5BK)}{n} \tag{5}$$

*Furthermore, if $\hat{\psi}_n$ is a data-dependent sub-root function with fixed point $\hat{r}^*$ such that*

$$\psi^*(r) > 2(10 \vee B)\mathbb{E}_\sigma \left[ \mathfrak{R}_n \{ f \in star(\mathcal{F}) \mid \mathcal{P}^n f^2 \leq 2r \} \right] + \frac{2(10 \vee B + 11)\delta}{n}$$

*then with probability at least $1 - 2e^\delta$, it holds that $\hat{r}^* \geq r^*$; as a consequence, Equations (4) and (5) hold with $r^*$ replaced by $\hat{r}^*$*

We now start the proof of Theorem 1. As discussed in the proof sketch in Section 5, we decompose the error into three terms, and bound each of them one by one.

$$\mathcal{E}(\mathrm{U}_{n,\lambda}, \mathrm{V}_{n,\lambda}) \quad = \underbrace{\langle \mathrm{U}_\mathrm{C}\mathrm{V}_\mathrm{C}^*, \mathrm{C} \rangle_{L(\mathcal{H}_\mathcal{Y}, \mathcal{H}_\mathcal{X})} - \left\langle \mathrm{U}_{\mathrm{C},\lambda}\mathrm{V}_{\mathrm{C},\lambda}^*, \mathrm{C}_\lambda \right\rangle_{L(\mathcal{H}_\mathcal{Y}, \mathcal{H}_\mathcal{X})}}_{\textbf{Error 1}}$$

$$+ \underbrace{\left\langle \mathrm{U}_{n,\lambda}\mathrm{V}_{n,\lambda}^*, \mathrm{C}_\lambda - \mathrm{C} \right\rangle_{L(\mathcal{H}_\mathcal{Y}, \mathcal{H}_\mathcal{X})}}_{\textbf{Error 2}}$$

$$+ \underbrace{\left\langle \mathrm{U}_{\mathrm{C},\lambda}\mathrm{V}_{\mathrm{C},\lambda}^* - \mathrm{U}_{n,\lambda}\mathrm{V}_{n,\lambda}^*, \mathrm{C}_\lambda \right\rangle_{L(\mathcal{H}_\mathcal{Y}, \mathcal{H}_\mathcal{X})}}_{\textbf{Error 3}}$$

The three error terms are bound in Lemmas 11, 14, and 15 respectively. Combining them gives the guarantee stated in Theorem 1. In the following subsections, we give the proofs of the afore-mentioned lemmas.

## B.1 Bounding the Error 1

**Lemma 11** (Error 1). *Error 1 is bounded as,*

$$\langle \mathrm{U}_\mathrm{C}\mathrm{V}_\mathrm{C}^*, \mathrm{C} \rangle_{L(\mathcal{H}_\mathcal{Y}, \mathcal{H}_\mathcal{X})} - \left\langle \mathrm{U}_{\mathrm{C},\lambda}\mathrm{V}_{\mathrm{C},\lambda}^*, \mathrm{C}_\lambda \right\rangle_{L(\mathcal{H}_\mathcal{Y}, \mathcal{H}_\mathcal{X})} \leq \frac{8k}{\sigma_k(\mathrm{C})} \left( \left( \sqrt{\lambda_\mathcal{X}} + \sqrt{\lambda_\mathcal{Y}} \right) + 2 \left( \sqrt{\lambda_\mathcal{X}} + \sqrt{\lambda_\mathcal{Y}} \right)^2 \right)$$

*Proof.* Note that $\mathrm{U}_\mathrm{C}$ and $\mathrm{V}_\mathrm{C}$ are operators corresponding to the first $k$ left and right singular functions of $\mathrm{C}$. Therefore $\langle \mathrm{U}_\mathrm{C}\mathrm{V}_\mathrm{C}^*, \mathrm{C} \rangle_{L(\mathcal{H}_\mathcal{Y}, \mathcal{H}_\mathcal{X})} - \left\langle \mathrm{U}_{\mathrm{C},\lambda}\mathrm{V}_{\mathrm{C},\lambda}^*, \mathrm{C}_\lambda \right\rangle_{L(\mathcal{H}_\mathcal{Y}, \mathcal{H}_\mathcal{X})}$ is difference of the sum of first $k$ singular values of $\mathrm{C}$ and $\mathrm{C}_\lambda$. That is, $\langle \mathrm{U}_\mathrm{C}\mathrm{V}_\mathrm{C}^*, \mathrm{C} \rangle_{L(\mathcal{H}_\mathcal{Y}, \mathcal{H}_\mathcal{X})} = \sum_{i=1}^k \sigma_i(\mathrm{C})$ and $\left\langle \mathrm{U}_{\mathrm{C},\lambda}\mathrm{V}_{\mathrm{C},\lambda}^*, \mathrm{C}_\lambda \right\rangle_{L(\mathcal{H}_\mathcal{Y}, \mathcal{H}_\mathcal{X})} = \sum_{i=1}^k \sigma_i(\mathrm{C}_\lambda)$. We

therefore have,

$$
\begin{aligned}
\left\langle U_C V_C^*, C\right\rangle_{L(\mathcal{H}_\mathcal{Y}, \mathcal{H}_\mathcal{X})} - \left\langle U_{C,\lambda} V_{C,\lambda}^*, C_\lambda\right\rangle_{L(\mathcal{H}_\mathcal{Y}, \mathcal{H}_\mathcal{X})} &= \sum_{i=1}^k \left(\sigma_i(C) - \sigma_i(C_\lambda)\right) \\
&= \sum_{i=1}^k \left(\sqrt{\lambda_i(CC^*)} - \sqrt{\lambda_i(C_\lambda C_\lambda^*)}\right) \\
&= \sum_{i=1}^k \frac{(\lambda_i(CC^*) - \lambda_i(C_\lambda C_\lambda^*))}{\sqrt{\lambda_i(CC^*)} + \sqrt{\lambda_i(C_\lambda C_\lambda^*)}} \\
&\leq \sum_{i=1}^k \frac{(\lambda_i(CC^*) - \lambda_i(C_\lambda C_\lambda^*))}{\sqrt{\lambda_i(CC^*)}}
\end{aligned}
$$

where the first inequality follows because $\lambda(C_\lambda C_\lambda^*) \geq 0$. We now apply perturbation bounds, in particular Weyl's inequality to bound the difference of eigenvalues of operators using the norm of their difference. We therefore get

$$
\begin{aligned}
&\left\langle U_C V_C^*, C\right\rangle_{L(\mathcal{H}_\mathcal{Y}, \mathcal{H}_\mathcal{X})} - \left\langle U_{C,\lambda} V_{C,\lambda}^*, C_\lambda\right\rangle_{L(\mathcal{H}_\mathcal{Y}, \mathcal{H}_\mathcal{X})} \\
&\leq \sum_{i=1}^k \frac{1}{\sigma_i(C)} \|CC^* - C_\lambda C_\lambda^*\| \\
&= \sum_{i=1}^k \frac{1}{\sigma_i(C)} \|CC^* - CC_\lambda^* + CC_\lambda^* - C_\lambda C_\lambda^*\| \\
&\leq \frac{k}{\sigma_k(C)} \left(\|C\| + \|C_\lambda\|\right) \|C - C_\lambda\| \\
&\leq \frac{k}{\sigma_k(C)} \left(2\|C\| + \|C - C_\lambda\|\right) \|C - C_\lambda\| \\
&\leq \frac{k}{\sigma_k(C)} \left(2\|C - C_\lambda\| + \|C - C_\lambda\|^2\right)
\end{aligned}
$$

where in the second inequality we used the fact that singular values are ordered, in the last inquality that $\|C\| \leq 1$ and triangle inequality in others. We now appeal to Lemma 12 that bounds $\|C - C_\lambda\|$ which completes the proof. $\qquad\square$

**Lemma 12.** *Given* $C = C_\mathcal{X}^{-1/2} C_{\mathcal{X}\mathcal{Y}} C_\mathcal{Y}^{-1/2}$ *and its regularized counterpart* $C_\lambda = (C_\mathcal{X} + \lambda_\mathcal{X} I_\mathcal{X})^{-1/2} C_{\mathcal{X}\mathcal{Y}} (C_\mathcal{Y} + \lambda_\mathcal{Y} I_\mathcal{Y})^{-1/2}$. *Assuming* $C$ *is compact, there difference is bounded as,*

$$
\|C - C_\lambda\| \leq 2(\sqrt{\lambda_\mathcal{X}} + \sqrt{\lambda_\mathcal{Y}})
$$

*Proof.* The proof follows the proof of Lemma 7 in Fukumizu et al. (2007).

$$
\begin{aligned}
&\|C_\lambda - C\| \\
=\ & \left\| (C_\mathcal{X} + \lambda_\mathcal{X} I_\mathcal{X})^{-1/2} C_{\mathcal{X}\mathcal{Y}} (C_\mathcal{Y} + \lambda_\mathcal{Y} I_\mathcal{Y})^{-1/2} - (C_\mathcal{X}^{-1/2} C_{\mathcal{X}\mathcal{Y}} C_\mathcal{Y}^{-1/2} \right\| \\
\leq\ & \left\| (C_\mathcal{X} + \lambda_\mathcal{X} I_\mathcal{X})^{-1/2} C_{\mathcal{X}\mathcal{Y}} (C_\mathcal{Y} + \lambda_\mathcal{Y} I_\mathcal{Y})^{-1/2} - C_\mathcal{X}^{-1/2} C_{\mathcal{X}\mathcal{Y}} (C_\mathcal{Y} + \lambda_\mathcal{Y} I_\mathcal{Y})^{-1/2} \right\| \\
& + \left\| (C_\mathcal{X})^{-1/2} C_{\mathcal{X}\mathcal{Y}} (C_\mathcal{Y} + \lambda_\mathcal{Y} I_\mathcal{Y})^{-1/2} - C_\mathcal{X}^{-1/2} C_{\mathcal{X}\mathcal{Y}} C_\mathcal{Y}^{-1/2} \right\| \\
=\ & \left\| ((C_\mathcal{X} + \lambda_\mathcal{X} I_\mathcal{X})^{-1/2} - C_\mathcal{X}^{-1/2}) C_{\mathcal{X}\mathcal{Y}} (C_\mathcal{Y} + \lambda_\mathcal{Y} I_\mathcal{Y})^{-1/2}) \right\| \\
& + \left\| C_\mathcal{X}^{-1/2} C_{\mathcal{X}\mathcal{Y}} ((C_\mathcal{Y} + \lambda_\mathcal{Y} I_\mathcal{Y})^{-1/2} - C_\mathcal{Y}^{-1/2}) \right\| \\
=\ & \left\| ((C_\mathcal{X} + \lambda_\mathcal{X} I_\mathcal{X})^{-1/2} C_\mathcal{X}^{1/2} - I_\mathcal{X}) C_\mathcal{X}^{-1/2} C_{\mathcal{X}\mathcal{Y}} (C_\mathcal{Y} + \lambda_\mathcal{Y} I_\mathcal{Y})^{-1/2}) \right\| \\
& + \left\| C(C_\mathcal{Y}^{1/2} (C_\mathcal{Y} + \lambda_\mathcal{Y} I_\mathcal{Y})^{-1/2} - I_\mathcal{Y}) \right\| \\
=\ & \left\| ((C_\mathcal{X} + \lambda_\mathcal{X} I_\mathcal{X})^{-1/2} C_\mathcal{X}^{1/2} - I_\mathcal{X}) C C_\mathcal{Y}^{1/2} (C_\mathcal{Y} + \lambda_\mathcal{Y} I_\mathcal{Y})^{-1/2} \right\| \\
& + \left\| C(C_\mathcal{Y}^{1/2} (C_\mathcal{Y} + \lambda_\mathcal{Y} I_\mathcal{Y})^{-1/2} - I_\mathcal{Y}) \right\| \\
\leq\ & \left\| ((C_\mathcal{X} + \lambda_\mathcal{X} I_\mathcal{X})^{-1/2} C_\mathcal{X}^{1/2} - I_\mathcal{X}) C \right\| + \left\| C(C_\mathcal{Y}^{1/2} (C_\mathcal{Y} + \lambda_\mathcal{Y} I_\mathcal{Y})^{-1/2} - I_\mathcal{Y}) \right\|
\end{aligned}
$$

where the last inequality follows since $\left\| C_\mathcal{Y}^{1/2} (C_\mathcal{Y} + \lambda_\mathcal{Y} I_\mathcal{Y})^{-1/2} \right\|_{L(\mathcal{H}_\mathcal{Y})} \leq 1$ for positive $\lambda_\mathcal{Y}$. We will now bound the two terms separately. Suppose for the first term, the operator norm is realized for $w \in \mathcal{H}_\mathcal{Y}$, $\|w\| = 1$ and $v = Cw$. Fukumizu et al. (2007) remarks that the range of $C$ is contained in the closure of the range of $C_\mathcal{X}$. That is, $v \in \{C\tilde{w} : \tilde{w} \in \mathcal{H}_\mathcal{Y}, \|\tilde{w}\| \leq 1\} \cap \text{Closure}(\text{Range}(C_\mathcal{X}))$. Since $C_\mathcal{X}$ is continuous, we have that for any $\epsilon > 0$, there exists $\tilde{v} \in \{C\tilde{w} : \tilde{w} \in \mathcal{H}_\mathcal{Y}, \|\tilde{w}\| \leq 1\} \cap \text{Range}(C_\mathcal{X})$ such that $\|v - \tilde{v}\|_{\mathcal{H}_\mathcal{X}} \leq \epsilon$. Further, there exists $\tilde{w} \in \mathcal{H}_\mathcal{Y}, \tilde{u} \in \mathcal{H}_\mathcal{X}$ such that $\tilde{v} = C\tilde{w} = C_\mathcal{X}\tilde{u}$. Plugging this in the first term, we get,

$$
\begin{aligned}
& \left\| ((C_\mathcal{X} + \lambda_\mathcal{X} I_\mathcal{X})^{-1/2} C_\mathcal{X}^{1/2} - I_\mathcal{X}) v \right\|_{\mathcal{H}_\mathcal{X}} \\
=\ & \left\| ((C_\mathcal{X} + \lambda_\mathcal{X} I_\mathcal{X})^{-1/2} C_\mathcal{X}^{1/2} - I_\mathcal{X}) (\tilde{v} + v - \tilde{v}) \right\|_{\mathcal{H}_\mathcal{X}} \\
\leq\ & \left\| ((C_\mathcal{X} + \lambda_\mathcal{X} I_\mathcal{X})^{-1/2} C_\mathcal{X}^{1/2} - I_\mathcal{X}) C_\mathcal{X}\tilde{u} \right\|_{\mathcal{H}_\mathcal{X}} + \left\| ((C_\mathcal{X} + \lambda_\mathcal{X} I_\mathcal{X})^{-1/2} C_\mathcal{X}^{1/2} - I_\mathcal{X}) (v - \tilde{v}) \right\|_{\mathcal{H}_\mathcal{X}} \\
\leq\ & \left\| (C_\mathcal{X} + \lambda_\mathcal{X} I_\mathcal{X})^{-1/2} C_\mathcal{X}^{1/2} ((C_\mathcal{X}^{1/2} - (C_\mathcal{X} + \lambda_\mathcal{X} I_\mathcal{X})^{1/2}) C_\mathcal{X}^{1/2}\tilde{u} \right\|_{\mathcal{H}_\mathcal{X}} + \left\| ((C_\mathcal{X} + \lambda_\mathcal{X} I_\mathcal{X})^{-1/2} C_\mathcal{X}^{1/2} - I_\mathcal{X}) \right\| \|v - \tilde{v}\|_{\mathcal{H}_\mathcal{X}} \\
\leq\ & \left\| (C_\mathcal{X} + \lambda_\mathcal{X} I_\mathcal{X})^{-1/2} C_\mathcal{X}^{1/2} \right\| \left\| C_\mathcal{X}^{1/2}\tilde{u} \right\|_{\mathcal{H}_\mathcal{X}} \left\| C_\mathcal{X}^{1/2} - (C_\mathcal{X} + \lambda_\mathcal{X} I_\mathcal{X})^{1/2} \right\| + \epsilon \\
\leq\ & 2 \left\| C_\mathcal{X}^{1/2}\tilde{u} \right\| \cdot \left\| C_\mathcal{X}^{1/2} - (C_\mathcal{X} + \lambda_\mathcal{X} I_\mathcal{X})^{1/2} \right\|
\end{aligned}
$$

where in the first inequality, we used the triangle inequality, where the second equality follows because $(C_\mathcal{X} + \lambda_\mathcal{X} I_\mathcal{X})^{-1/2} C_\mathcal{X}^{1/2} (C_\mathcal{X} + \lambda_\mathcal{X} I_\mathcal{X})^{1/2} C_\mathcal{X}^{1/2} = C_\mathcal{X}$ since these operators are commutative i.e. have the same eigenspaces. In the second to last inequality, we use that $\left\| ((C_\mathcal{X} + \lambda_\mathcal{X} I_\mathcal{X})^{-1/2} C_\mathcal{X}^{1/2} - I_\mathcal{X}) \right\| \leq 1$ which follows because $\left\| ((C_\mathcal{X} + \lambda_\mathcal{X} I_\mathcal{X})^{-1/2} C_\mathcal{X}^{1/2} - I_\mathcal{X}) \right\| = \max_i \left| \frac{\sqrt{\lambda_i(C_\mathcal{X})}}{\sqrt{\lambda_i(C_\mathcal{X}) + \lambda_\mathcal{X}}} - 1 \right| = \max_i \left| \frac{\sqrt{\lambda_i(C_\mathcal{X})} - \sqrt{\lambda_i(C_\mathcal{X}) + \lambda_\mathcal{X}}}{\sqrt{\lambda_i(C_\mathcal{X}) + \lambda_\mathcal{X}}} \right| \leq 1$ as $C_\mathcal{X}$ is trace-class so its eigenvalues go to 0. Finally, the last step follows from $\left\| C_\mathcal{X}^{1/2} (C_\mathcal{X} + \lambda_\mathcal{X} I_\mathcal{X})^{-1/2} \right\| \leq 1$ for positive $\lambda_\mathcal{X}$ and by choosing $\epsilon = \left\| (C_\mathcal{X} + \lambda_\mathcal{X} I_\mathcal{X})^{-1/2} C_\mathcal{X}^{1/2} \right\| \left\| C_\mathcal{X}^{1/2}\tilde{u} \right\|_{\mathcal{H}_\mathcal{X}} \left\| C_\mathcal{X}^{1/2} - (C_\mathcal{X} + \lambda_\mathcal{X} I_\mathcal{X})^{1/2} \right\|$. We will now argue that $\left\| C_\mathcal{X}^{1/2}\tilde{u} \right\| \leq 1$. This follows by our assumption 1 that the operator $C_\mathcal{X}^{-1} C_{\mathcal{X}\mathcal{Y}} C_\mathcal{Y}^{-1/2}$

is Hilbert-Schmidt. With some abuse of notation, we have that $\left\|C_{\mathcal{X}}^{1/2}\tilde{u}\right\| = \left\|C_{\mathcal{X}}^{-1/2}\tilde{v}\right\| = \left\|C_{\mathcal{X}}^{-1/2}C\tilde{w}\right\| = \left\|C_{\mathcal{X}}^{-1}C_{\mathcal{X}\mathcal{Y}}C_{\mathcal{Y}}^{-1/2}\tilde{w}\right\| \leq \left\|C_{\mathcal{X}}^{-1}C_{\mathcal{X}\mathcal{Y}}C_{\mathcal{Y}}^{-1/2}\right\| \|\tilde{w}\| \leq 1$ where the first term is bounded by 1 by assumption, and the second because $\|\tilde{w}\| \leq 1$. Similarly, using the same argument for the other term, we get,

$$\left\|C(C_{\mathcal{Y}}^{1/2}(C_{\mathcal{Y}} + \lambda_{\mathcal{Y}}I_{\mathcal{Y}})^{-1/2} - I_{\mathcal{Y}})\right\| \leq 2\left\|C_{\mathcal{Y}}^{1/2}\tilde{u}\right\|\left\|C_{\mathcal{Y}}^{1/2} - (C_{\mathcal{Y}} + \lambda_{\mathcal{Y}}I_{\mathcal{Y}})^{1/2})\right\|$$
$$\leq 2\left\|C_{\mathcal{Y}}^{1/2} - (C_{\mathcal{Y}} + \lambda_{\mathcal{Y}}I_{\mathcal{Y}})^{1/2})\right\|$$

Finally using Lemma 13, we get the bound $2(\sqrt{\lambda_{\mathcal{X}}} + \sqrt{\lambda_{\mathcal{Y}}})$. $\qquad\square$

**Lemma 13.** *For self-adjoint trace-class operator $C_{\mathcal{X}}$ and positive $\lambda_{\mathcal{X}}$*

$$\left\|C_{\mathcal{X}}^{1/2} - (C_{\mathcal{X}} + \lambda_{\mathcal{X}}I_{\mathcal{X}})^{1/2}\right\| \leq \sqrt{\lambda_{\mathcal{X}}}$$

*Proof.* Since these operators are commutative,

$$\left\|C_{\mathcal{X}}^{1/2} - (C_{\mathcal{X}} + \lambda_{\mathcal{X}}I_{\mathcal{X}})^{1/2}\right\| = \max_i |\sqrt{\lambda_i(C_{\mathcal{X}})} - \sqrt{\lambda_i(C_{\mathcal{X}} + \lambda_{\mathcal{X}}I_{\mathcal{X}})}| \leq \sqrt{\lambda_{\mathcal{X}}}$$

since the operator being trace class implies that the eigenvalues go to 0. $\qquad\square$

## B.2 Bounding the Error 2

**Lemma 14** (Error 2). *Error 2 is bounded as*

$$\langle U_{n,\lambda}V_{n,\lambda}^*, C - C_\lambda \rangle \leq k\|C_\lambda - C\| \leq 4k\left(\sqrt{\lambda_{\mathcal{X}}} + \sqrt{\lambda_{\mathcal{Y}}}\right)$$

*Proof.* The first inequality simply follows from Holder's inequality with conjugates Schatten 1 and $\infty$ norms, and the second using Lemma 12. $\qquad\square$

## B.3 Bounding the Error 3

**Lemma 15** (Error 3). *With probability at least $1 - \delta$, Error 3 is bounded as,*

$$\langle U_{C,\lambda}V_{C,\lambda}^* - U_{n,\lambda}V_{n,\lambda}^*, C_\lambda \rangle_{L(\mathcal{H}_{\mathcal{Y}}, \mathcal{H}_{\mathcal{X}})} \leq \inf_{h\geq 0}\left\{\frac{12\alpha_\rho h}{\lambda_{\mathcal{X}}\lambda_{\mathcal{Y}}n} + \frac{24\alpha_\rho\sqrt{h}}{\lambda_{\mathcal{X}}\lambda_{\mathcal{Y}}n} + \frac{24}{\sqrt{n}}\sqrt{\frac{k}{\lambda_{\mathcal{X}}\lambda_{\mathcal{Y}}}\sum_{j>h}\lambda_j(\mathfrak{C}')}\right\}$$
$$+ \frac{12\alpha_\rho}{\lambda_{\mathcal{X}}\lambda_{\mathcal{Y}}n} + \frac{22\beta\sqrt{k}\log(1/\delta)}{\sqrt{\lambda_{\mathcal{X}}\lambda_{\mathcal{Y}}}n} + \frac{10\alpha_\rho\log(1/\delta)}{\lambda_{\mathcal{X}}\lambda_{\mathcal{Y}}n}$$

*Proof.* The proof follows the application of local Rademacher complexity analysis technique for Kernel PCA Blanchard et al. (2007), with modifications arising from differences in the problems. We start with the function class

$$\mathcal{F} = \{f_{U,V} : (x, y) \to \left\langle \overline{U}_{C,\lambda}\overline{V}_{C,\lambda}^* - UV^*, C_{x,y}\right\rangle_{L(\mathcal{H}_{\mathcal{Y}}, \mathcal{H}_{\mathcal{X}})} \mid U^*C_{\mathcal{X}}^{\lambda_{\mathcal{X}}}U = I_k, V^*C_{\mathcal{Y}}^{\lambda_{\mathcal{Y}}}V = I_k\}$$

where $\overline{U}_{C,\lambda} = \left(C_{\mathcal{X}}^{\lambda_{\mathcal{X}}}\right)^{1/2}U_C, \overline{V}_{C,\lambda} = \left(C_{\mathcal{Y}}^{\lambda_{\mathcal{Y}}}\right)^{1/2}V_C$ and $C_{x,y} = x \otimes_{L(\mathcal{H}_{\mathcal{Y}}, \mathcal{H}_{\mathcal{X}})} y$

We look at the function class $\mathcal{G} = \tau\mathcal{F}$, where $\tau = \frac{\sqrt{\lambda_{\mathcal{X}}\lambda_{\mathcal{Y}}}}{2\beta\sqrt{k}}$. From Lemma 17, we get that for $f \in \mathcal{G}$ the range of $f$ is contained in $[-1, 1]$. We then show in Lemma 22 that $\mathbb{E}\left[f^2\right] \leq \mu\mathbb{E}\left[f\right]$ where $\mu = \frac{2\alpha_\rho\tau}{\lambda_{\mathcal{X}}\lambda_{\mathcal{Y}}}$ and $\alpha_\rho = \frac{\mathbb{E}_{x,y,x',y'}\left[\langle x,x'\rangle_{\mathcal{H}_{\mathcal{X}}}^2 \langle y,y'\rangle_{\mathcal{H}_{\mathcal{Y}}}^2\right]}{\lambda_k(C) - \lambda_{k+1}(C)}$ for $f \in \mathcal{G}$.

From Lemma 16, we have that , $\|UV^*\|^2_{L(\mathcal{H_Y,H_X})} \leq \frac{k}{\lambda_\mathcal{X}\lambda_\mathcal{Y}}$. Similarly, we also have $\left\|\overline{U}_{C,\lambda}\overline{V}_{C,\lambda}\right\|^2_{L(\mathcal{H_Y,H_X})} \leq \frac{k}{\lambda_\mathcal{X}\lambda_\mathcal{Y}}$. Therefore, we have

$$\left\|\overline{U}_C\overline{V}_C^* - UV^*\right\|^2_{L(\mathcal{H_Y,H_X})}$$
$$\leq 2\left(\left\|\overline{U}_C\overline{V}_C^*\right\|^2_{L(\mathcal{H_Y,H_X})} + \|UV^*\|^2_{L(\mathcal{H_Y,H_X})}\right)$$
$$\leq \frac{4k}{\lambda_\mathcal{X}\lambda_\mathcal{Y}}$$

Therefore, we can write,

$$\mathcal{F} \subseteq \left\{(x,y) \to \langle\Gamma, C_{x,y}\rangle_{L(\mathcal{H_Y,H_X})} \mid \Gamma \in L(\mathcal{H_Y,H_X}), \|\Gamma\|^2_{L(\mathcal{H_Y,H_X})} \leq \frac{4k}{\lambda_\mathcal{X}\lambda_\mathcal{Y}}\right\} =: \mathcal{H}$$

We will now concern ourselves with the set $\mathcal{H}$. We have

$$\mathbb{E}\left[f^2\right] = \mathbb{E}\left[\langle\Gamma, C_{xy}\rangle^2_{L(\mathcal{H_Y,H_X})}\right] = \mathbb{E}_{x,y}\left[\langle\Gamma, C_{x,y} \otimes_{L(\mathcal{H_Y,H_X})} C_{x,y}\Gamma\rangle_{L(\mathcal{H_Y,H_X})}\right] = \langle\Gamma, \mathfrak{C}\Gamma\rangle_{L(\mathcal{H_Y,H_X})}$$

where $\mathfrak{C} \in L(L(\mathcal{H_Y,H_X}))$ is defined as $\mathfrak{C} := \mathbb{E}_{x,y}\left[C_{x,y} \otimes_{L(\mathcal{H_Y,H_X})} C_{x,y}\right]$. Since the set $\mathcal{F}$ is contained in $\mathcal{H}$, which is a convex set and contains origin, $\text{star}(\mathcal{F}_k)$ is also contained in $\mathcal{H}$.

$$\text{star}(\mathcal{F}) \subseteq \left\{(x,y) \to \langle\Gamma, C_{x,y}\rangle_{L(\mathcal{H_Y,H_X})} \mid \Gamma \in L(\mathcal{H_Y,H_X}), \|\Gamma\|^2_{L(\mathcal{H_Y,H_X})} \leq \frac{4k}{\lambda_\mathcal{X}\lambda_\mathcal{Y}}\right\}$$

Moreover,

$$\left\{g \in \text{star}(\mathcal{G}) \mid \mathbb{E}\left[g^2\right] \leq r\right\} = \tau\left\{g \in \text{star}(\mathcal{F}) \mid \mathbb{E}\left[g^2\right] \leq \tau^{-2}r\right\}$$
$$\subset \tau\{(x,y) \to \langle\Gamma, C_{x,y}\rangle_{L(\mathcal{H_Y,H_X})} \mid \Gamma \in L(\mathcal{H_Y,H_X}),$$
$$\|\Gamma\|^2_{L(\mathcal{H_Y,H_X})} \leq \frac{4k}{\lambda_\mathcal{X}\lambda_\mathcal{Y}}, \langle\Gamma, \mathfrak{C}\Gamma\rangle_{L(\mathcal{H_Y,H_X})} \leq \tau^{-2}r\}$$
$$=: \mathcal{S}_r$$

We now want to bound the Rademacher Complexity of $\mathcal{S}_r$ which is $\mathfrak{R}_n(\mathcal{S}_r) = \mathbb{E}_{x,y}\mathbb{E}_\sigma\left[\frac{1}{n}\sup_{f\in S_r}\sum_{i=1}^n \sigma_i f(x_i, y_i)\right]$.

From Lemma 18, we get that the Rademacher complexity of $\mathcal{S}_r$ is bounded as follows,

$$\mathfrak{R}_n(\mathcal{S}_r) \leq \sqrt{\frac{r}{n}} + \frac{1}{\sqrt{n}}\inf_{h\geq 0}\left(\sqrt{r}h + 2\tau\sqrt{\frac{k}{\lambda_\mathcal{X}\lambda_\mathcal{Y}}\sum_{j>h}\lambda_j(\mathfrak{C}')}\right) =: \frac{\psi(r)}{\mu}$$

Note that this is a sub-root function in $r$, as the infimum for sub-root functions is sub-root. We now need to upper bound the fixed point of $\psi(r)$. Define $\xi := \frac{\mu}{\tau} = \frac{2\alpha_\rho}{\lambda_\mathcal{X}\lambda_\mathcal{Y}}$. We want,

$$r^* = \psi(r^*) = \frac{\xi\tau}{\sqrt{n}}\left(\sqrt{r^*}\left(\sqrt{h}+1\right) + 2\tau\sqrt{\frac{k}{\lambda_\mathcal{X}\lambda_\mathcal{Y}}\sum_{j>h}\lambda_j(\mathfrak{C}')}\right)$$

From Lemma 21, we have that fixed point $r^*$ is bounded as

$$r^* \leq \tau^2\left(\inf_{h\geq 0}\left\{\frac{\xi^2 h}{n} + \frac{2\xi^2\sqrt{h}}{n} + \frac{4\xi}{\sqrt{n}}\sqrt{\frac{k}{\lambda_\mathcal{X}\lambda_\mathcal{Y}}\sum_{j>h}\lambda_j(\mathfrak{C}')}\right\} + \frac{\xi^2}{n}\right)$$

Let

$$\kappa(\xi, n) = \inf_{h \geq 0} \left\{ \frac{\xi^2 h}{n} + \frac{2\xi^2 \sqrt{h}}{n} + \frac{4\xi}{\sqrt{n}} \sqrt{\frac{k}{\lambda_{\mathcal{X}} \lambda_{\mathcal{Y}}} \sum_{j > h} \lambda_j(\mathfrak{C}')} \right\}$$

Having shown that it satisfies all the conditions of Theorem 10, we now apply the theorem. We get that for any $K > 1$, with probability at least $1 - \delta$, $\forall\, U \in L(\mathcal{H}_{\mathcal{X}}, \mathbb{R}^k), V \in L(\mathcal{H}_{\mathcal{Y}}, \mathbb{R}^k)$ in the feasible set, we have,

$$\mathbb{E}\left[\tau f_{U,V}\right] \leq \frac{K \mathbb{E}_n f_{U,V}}{K - 1} + \frac{6K r^*}{\tau \xi} + \frac{(11 + 5\tau \xi K) \log \delta}{n}$$

$$\leq \frac{K \mathbb{E}_n f_{U,V}}{K - 1} + \frac{6K \kappa(\xi, k, n) \tau}{\xi} + \frac{6K \xi \tau}{n} + \frac{11 \log \delta}{n} + \frac{5\xi K \log \delta \tau}{n}$$

Therefore,

$$\mathbb{E}\left[f_{U,V}\right] \leq \frac{K \mathbb{E}_n f_{U,V}}{K - 1} + \frac{6K \kappa(\xi, n)}{\xi} + \frac{6K \xi}{n} + \frac{11 \log \delta}{\tau n} + \frac{5\xi K \log \delta}{n}$$

We set $U = C_{\mathcal{X}}^{-1/2} U_n$ and $V = C_{\mathcal{Y}}^{-1/2} V_n$, therefore we get $\mathbb{E}_n f_{U,V} \leq 0$. where $\mathbb{E}_n$ is the expectation with respect to the empirical measure. Letting $K \to 1$, we get, with probability at least $1 - \delta$,

$$\left\langle U_{C,\lambda} V_{C,\lambda}^* - U_{n,\lambda} V_{n,\lambda}^*, C_\lambda \right\rangle_{L(\mathcal{H}_{\mathcal{Y}}, \mathcal{H}_{\mathcal{X}})}$$

$$\leq \frac{6\kappa(\xi, n)}{\xi} + \frac{6\xi}{n} + \frac{11 \log \delta}{\tau n} + \frac{5\xi \log \delta}{n}$$

Note that $\xi = \frac{2\alpha_\rho}{\lambda_{\mathcal{X}} \lambda_{\mathcal{Y}}}$, where $\alpha_\rho = \frac{\mathbb{E}_{x,y,x',y'}\left[\langle x, x' \rangle_{\mathcal{H}_{\mathcal{X}}}^2 \langle y, y' \rangle_{\mathcal{H}_{\mathcal{Y}}}^2\right]}{(\lambda_k(C) - \lambda_{k+1}(C))}$, and $\tau = \frac{\sqrt{\lambda_{\mathcal{X}} \lambda_{\mathcal{Y}}}}{2\beta \sqrt{k}}$. We now substitute these to obtain the final bound. We have,

$$\frac{\kappa(\xi, k, n)}{\xi} = \frac{1}{\xi} \inf_{h \geq 0} \left\{ \frac{\xi^2 h}{n} + \frac{2\xi^2 \sqrt{h}}{n} + \frac{4\xi}{\sqrt{n}} \sqrt{\frac{k}{\lambda_{\mathcal{X}} \lambda_{\mathcal{Y}}} \sum_{j > h} \lambda_j(\mathfrak{C}')} \right\}$$

$$= \inf_{h \geq 0} \left\{ \frac{\xi h}{n} + \frac{2\xi \sqrt{h}}{n} + \frac{4}{\sqrt{n}} \sqrt{\frac{k}{\lambda_{\mathcal{X}} \lambda_{\mathcal{Y}}} \sum_{j > h} \lambda_j(\mathfrak{C}')} \right\}$$

$$= \inf_{h \geq 0} \left\{ \frac{2\alpha_\rho h}{\lambda_{\mathcal{X}} \lambda_{\mathcal{Y}} m} + \frac{4\alpha_\rho \sqrt{h}}{\lambda_{\mathcal{X}} \lambda_{\mathcal{Y}} n} + \frac{4}{\sqrt{n}} \sqrt{\frac{k}{\lambda_{\mathcal{X}} \lambda_{\mathcal{Y}}} \sum_{j > h} \lambda_j(\mathfrak{C}')} \right\}$$

Therefore, we get,

$$\left\langle U_{C,\lambda} V_{C,\lambda}^* - U_{n,\lambda} V_{n,\lambda}^*, C_\lambda \right\rangle_{L(\mathcal{H}_{\mathcal{Y}}, \mathcal{H}_{\mathcal{X}})}$$

$$\leq \inf_{h \geq 0} \left\{ \frac{12\alpha_\rho h}{\lambda_{\mathcal{X}} \lambda_{\mathcal{Y}} n} + \frac{24\alpha_\rho \sqrt{h}}{\lambda_{\mathcal{X}} \lambda_{\mathcal{Y}} n} + \frac{24}{\sqrt{n}} \sqrt{\frac{k}{\lambda_{\mathcal{X}} \lambda_{\mathcal{Y}}} \sum_{j > h} \lambda_j(\mathfrak{C}')} \right\} + \frac{12\alpha_\rho}{\lambda_{\mathcal{X}} \lambda_{\mathcal{Y}} n} + \frac{22\beta \sqrt{k} \log \delta}{\sqrt{\lambda_{\mathcal{X}} \lambda_{\mathcal{Y}}} n} + \frac{10\alpha_\rho \log \delta}{\lambda_{\mathcal{X}} \lambda_{\mathcal{Y}} n}$$

$$\square$$

**Lemma 16.** *If* $U$ *and* $V$ *satisfies* $U^* C_{\mathcal{X}}^{\lambda_{\mathcal{X}}} U = I_k$ *and* $V^* C_{\mathcal{Y}}^{\lambda_{\mathcal{Y}}} V = I_k$, *where* $C_{\mathcal{X}}^{\lambda_{\mathcal{X}}} = C_{\mathcal{X}} + \lambda_{\mathcal{X}} I_{\mathcal{X}}$ *and* $C_{\mathcal{Y}}^{\lambda_{\mathcal{Y}}} = C_{\mathcal{Y}} + \lambda_{\mathcal{Y}} I_{\mathcal{Y}}$, *then,*

$$\|UV^*\|_{L(\mathcal{H}_{\mathcal{Y}}, \mathcal{H}_{\mathcal{X}})}^2 \leq \frac{k}{\lambda_{\mathcal{X}} \lambda_{\mathcal{Y}}}$$

*Proof.* We have,

$$k = \left\| U^* C_{\mathcal{X}}^{\lambda_{\mathcal{X}}} U \right\|_{L(L(\mathcal{H}_{\mathcal{X}}, \mathbb{R}^k))}^2$$
$$\geq \lambda_{\mathcal{X}}^2 \left\| U^* U \right\|_{L(L(\mathcal{H}_{\mathcal{X}}, \mathbb{R}^k))}^2$$

Therefore, we get $\| U^* U \|_{L(L(\mathcal{H}_{\mathcal{X}}, \mathbb{R}^k))} \leq \frac{\sqrt{k}}{\lambda_{\mathcal{X}}}$. Similarly, we can show that $\| V^* V \|_{L(L(\mathcal{H}_{\mathcal{Y}}, \mathbb{R}^k))} \leq \frac{\sqrt{k}}{\lambda_{\mathcal{Y}}}$.

$$\| UV^* \|_{L(\mathcal{H}_{\mathcal{Y}}, \mathcal{H}_{\mathcal{X}})}^2 = \langle UV^*, UV^* \rangle_{L(\mathcal{H}_{\mathcal{Y}}, \mathcal{H}_{\mathcal{X}})}$$
$$= \langle U^* U, V^* V \rangle_{L(\mathbb{R}^k)}$$
$$\leq \| U^* U \|_{L(\mathbb{R}^k)} \| V^* V \|_{L(\mathbb{R}^k)}$$
$$\leq \frac{k}{\lambda_{\mathcal{X}} \lambda_{\mathcal{Y}}}$$

where we use the definition of adjoints in the second step and Cauchy-Schwartz inequality in the third step. $\square$

**Lemma 17.** *For any $f \in \mathcal{G} = \tau \mathcal{F}$ the range of $f$ is contained in $[-1, 1]$, where $\tau = \frac{\sqrt{\lambda_{\mathcal{X}} \lambda_{\mathcal{Y}}}}{2\beta\sqrt{k}}$*

*Proof.* From Lemma 16, we get that $\| UV^* \|_{L(\mathcal{H}_{\mathcal{Y}}, \mathcal{H}_{\mathcal{X}})}^2 \leq \frac{k}{\lambda_{\mathcal{X}} \lambda_{\mathcal{Y}}}$. Similarly, we also get $\left\| U_{C,\lambda} V_{C,\lambda}^* \right\|_{L(\mathcal{H}_{\mathcal{Y}}, \mathcal{H}_{\mathcal{X}})}^2 \leq \frac{k}{\lambda_{\mathcal{X}} \lambda_{\mathcal{Y}}}$.

We now note that,

$$\left\langle \overline{U}_{C,\lambda} \overline{V}_{C,,\lambda}^* - UV^*, C_{x,y} \right\rangle_{L(\mathcal{H}_{\mathcal{Y}}, \mathcal{H}_{\mathcal{X}})}^2 \leq \left\| \overline{U}_{C,\lambda} \overline{V}_{C,\lambda}^* - UV \right\|_{L(\mathcal{H}_{\mathcal{Y}}, \mathcal{H}_{\mathcal{X}})}^2 \| C_{x,y} \|_{L(\mathcal{H}_{\mathcal{Y}}, \mathcal{H}_{\mathcal{X}})}^2$$
$$\leq 2\beta^2 \left( \left\| \overline{U}_{C,\lambda} \overline{V}_{C,\lambda}^* \right\|_{L(\mathcal{H}_{\mathcal{Y}}, \mathcal{H}_{\mathcal{X}})}^2 + \| UV^* \|_{L(\mathcal{H}_{\mathcal{Y}}, \mathcal{H}_{\mathcal{X}})}^2 \right)$$
$$\leq \frac{4\beta^2 k}{\lambda_{\mathcal{X}} \lambda_{\mathcal{Y}}}$$

where in the first step we applied Cauchy-Schwartz inequality and in the second step, we used that $\| C_{x,y} \|_{L(\mathcal{H}_{\mathcal{Y}}, \mathcal{H}_{\mathcal{X}})} = \left\| x \otimes_{L(\mathcal{H}_{\mathcal{Y}}, \mathcal{H}_{\mathcal{X}})} y \right\|_{L(\mathcal{H}_{\mathcal{Y}}, \mathcal{H}_{\mathcal{X}})} \leq \| x \|_{\mathcal{H}_{\mathcal{X}}} \| y \|_{\mathcal{H}_{\mathcal{Y}}} \leq \beta^2$. Therefore, we have, $\left\langle \overline{U}_{C,\lambda} \overline{V}_{C,\lambda}^* - UV^*, C_{x,y} \right\rangle \leq \frac{2\beta\sqrt{k}}{\sqrt{\lambda_{\mathcal{X}} \lambda_{\mathcal{Y}}}}$.

Therefore, since any $\bar{f} \in \mathcal{F}$ range of $\bar{f} \leq \tau^{-1}$, we have for $f = \tau f \in \tau \mathcal{F} = \mathcal{G}$ its range $\leq 1$. The lower bound holds because $\overline{U}_C$ and $\overline{V}_C$ correspond to the optimal solution, therefore for any function $\bar{f} \in \mathcal{F}, \bar{f} \geq 0$. So, any $f \in \tau \mathcal{F} = \mathcal{G}, f \geq 0 \geq -1$. $\square$

**Lemma 18.** *The Rademacher complexity of $\mathcal{S}_r$, defined as,*

$$\mathcal{S}_r = \tau \{ (x, y) \to \langle \Gamma, C_{x,y} \rangle_{L(\mathcal{H}_{\mathcal{Y}}, \mathcal{H}_{\mathcal{X}})} \mid \Gamma \in L(\mathcal{H}_{\mathcal{Y}}, \mathcal{H}_{\mathcal{X}}), \| \Gamma \|_{L(\mathcal{H}_{\mathcal{Y}}, \mathcal{H}_{\mathcal{X}})}^2 \leq \frac{4k}{\lambda_{\mathcal{X}} \lambda_{\mathcal{Y}}}, \langle \Gamma, \mathfrak{C}\Gamma \rangle_{L(\mathcal{H}_{\mathcal{Y}}, \mathcal{H}_{\mathcal{X}})} \leq \tau^{-2} r \}$$

*is bounded as follows,*

$$\mathfrak{R}_n(\mathcal{S}_r) \leq \sqrt{\frac{r}{n}} + \frac{1}{\sqrt{n}} \inf_{h \geq 0} \left( \sqrt{r} h + 2\tau \sqrt{\frac{k}{\lambda_{\mathcal{X}} \lambda_{\mathcal{Y}}} \sum_{j > h} \lambda_j(\mathfrak{C}')} \right)$$

*Proof.* Note that we can write $\langle \Gamma, C_{x,y} \rangle_{L(\mathcal{H}_\mathcal{Y}, \mathcal{H}_\mathcal{X})} = \langle \Gamma, C_{x,y} - C_{\mathcal{X}\mathcal{Y}} \rangle_{L(\mathcal{H}_\mathcal{Y}, \mathcal{H}_\mathcal{X})} + \langle \Gamma, C_{\mathcal{X}\mathcal{Y}} \rangle_{L(\mathcal{H}_\mathcal{Y}, \mathcal{H}_\mathcal{X})}$. Equivalently, we can decompose the function class $\mathcal{S}_r$ into two classes $\mathcal{P}_r$ and $\mathcal{Q}_r$, defined as,

$$\mathcal{P}_r = \tau \left\{ (x,y) \to \langle \Gamma, C_{\mathcal{X}\mathcal{Y}} \rangle_{L(\mathcal{H}_\mathcal{Y}, \mathcal{H}_\mathcal{X})} \mid \langle \Gamma, \mathfrak{C}\Gamma \rangle_{L(\mathcal{H}_\mathcal{Y}, \mathcal{H}_\mathcal{X})} \leq \tau^{-2} r \right\}$$

and

$$\mathcal{Q}_r = \tau \Big\{ (x,y) \to \langle \Gamma, C_{x,y} - C_{\mathcal{X}\mathcal{Y}} \rangle_{L(\mathcal{H}_\mathcal{Y}, \mathcal{H}_\mathcal{X})} \mid \|\Gamma\|^2_{L(\mathcal{H}_\mathcal{Y}, \mathcal{H}_\mathcal{X})} \leq \frac{4k}{\lambda_\mathcal{X} \lambda_\mathcal{Y}},$$
$$\langle \Gamma, (\mathfrak{C} - C_{\mathcal{X}\mathcal{Y}} \otimes_{L(L(\mathcal{H}_\mathcal{Y}, \mathcal{H}_\mathcal{X}))} C_{\mathcal{X}\mathcal{Y}}) \Gamma \rangle_{L(\mathcal{H}_\mathcal{Y}, \mathcal{H}_\mathcal{X})} \leq \tau^{-2} r \Big\}$$

By a simple application of triangle inequality, we have,

$$\mathfrak{R}_n(\mathcal{S}_r) \leq \mathfrak{R}_n(\mathcal{P}_r) + \mathfrak{R}_n(\mathcal{Q}_r)$$

We bound the Rademacher complexities of the sets in Lemma 19 and 20 respectively. From them, we get,

$$\mathfrak{R}_n(\mathcal{P}_r) \leq \sqrt{\frac{r}{n}}$$

and

$$\mathfrak{R}_n(\mathcal{Q}_r) \leq \frac{1}{\sqrt{n}} \inf_{h \geq 0} \left( \sqrt{r} h + 2\tau \sqrt{\frac{k}{\lambda_\mathcal{X} \lambda_\mathcal{Y}} \sum_{j > h} \lambda_j(\mathfrak{C}')} \right)$$

Combining these, we have,

$$\mathfrak{R}_n(\mathcal{S}_r) \leq \sqrt{\frac{r}{n}} + \frac{1}{\sqrt{n}} \inf_{h \geq 0} \left( \sqrt{r} h + 2\tau \sqrt{\frac{k}{\lambda_\mathcal{X} \lambda_\mathcal{Y}} \sum_{j > h} \lambda_j(\mathfrak{C}')} \right)$$

$\square$

**Lemma 19.** *The Rademacher complexity of the set $\mathcal{P}_r$, defined as,*

$$\mathcal{P}_r = \tau \left\{ (x,y) \to \langle \Gamma, C_{\mathcal{X}\mathcal{Y}} \rangle_{L(\mathcal{H}_\mathcal{Y}, \mathcal{H}_\mathcal{X})} \mid \langle \Gamma, \mathfrak{C}\Gamma \rangle_{L(\mathcal{H}_\mathcal{Y}, \mathcal{H}_\mathcal{X})} \leq \tau^{-2} r \right\}$$

*is bounded as,*

$$\mathfrak{R}_n(\mathcal{P}_r) \leq \sqrt{\frac{r}{n}}$$

*Proof.* Since $\mathcal{P}_r$ contains only constant functions, we can easily bound its Rademacher complexity. In particular, for a set of scalars $Z \subset \mathbb{R}$, we have

$$\mathbb{E} \left[ \sup_{z \in Z} \left( z \sum_{i=1}^n \sigma_i \right) \right] = \left( \frac{\sup Z - \inf Z}{2} \right) \mathbb{E} \left[ \left| \sum_{i=1}^n \sigma_i \right| \right]$$
$$\leq \left( \frac{\sup Z - \inf Z}{2} \right) \sum_{i=1}^n \mathbb{E} \left[ |\sigma_i| \right]$$
$$= \frac{(\sup Z - \inf Z)\sqrt{n}}{2}$$

where the second step follows from Jensen's inequality, and last step from the fact that $\mathbb{E}\left[|\sigma|\right] = 1$ for a Rademacher random variable $\sigma$. Let $\bar{f} \in \mathbb{R}^m$ where each co-ordinate is the value of the constant function. Therefore, we have

$$
\begin{aligned}
\mathfrak{R}_n(\mathcal{P}_r) &= \frac{1}{n}\mathbb{E}_\sigma\left[\sup_{f\in\mathcal{P}_r}\sum_{i=1}^m\sigma_i\bar{f}\right] \\
&\leq \frac{1}{n}\cdot\frac{2\sqrt{n}}{2}\cdot\sup\left\{\langle\Gamma,\mathrm{C}_{\mathcal{X}\mathcal{Y}}\rangle_{L(\mathcal{H}_{\mathcal{Y}},\mathcal{H}_{\mathcal{X}})}\,|\,\langle\Gamma,\mathfrak{C}\Gamma\rangle_{L(\mathcal{H}_{\mathcal{Y}},\mathcal{H}_{\mathcal{X}})}\leq\tau^{-2}r\right\} \\
&\leq \sqrt{\frac{r}{n}}
\end{aligned}
$$

This follows because

$$
\langle\Gamma,\mathfrak{C}\Gamma\rangle_{L(\mathcal{H}_{\mathcal{Y}},\mathcal{H}_{\mathcal{X}})} = \mathbb{E}\left[\langle\mathrm{C}_{x,y},\Gamma\rangle^2_{L(\mathcal{H}_{\mathcal{Y}},\mathcal{H}_{\mathcal{X}})}\right] \geq \left(\mathbb{E}\left[\langle\mathrm{C}_{x,y},\Gamma\rangle_{L(\mathcal{H}_{\mathcal{Y}},\mathcal{H}_{\mathcal{X}})}\right]\right)^2
$$

from Jensen's inequality. $\qquad\square$

**Lemma 20.** *The Rademacher complexity of the set $\mathcal{Q}_r$, defined as*

$$
\begin{aligned}
\mathcal{Q}_r = \tau\Big\{(x,y)\to\langle\Gamma,\mathrm{C}_{x,y}-\mathrm{C}_{\mathcal{X}\mathcal{Y}}\rangle_{L(\mathcal{H}_{\mathcal{Y}},\mathcal{H}_{\mathcal{X}})}\;\big|\;\|\Gamma\|^2_{L(\mathcal{H}_{\mathcal{Y}},\mathcal{H}_{\mathcal{X}})}\leq\frac{4k}{\lambda_{\mathcal{X}}\lambda_{\mathcal{Y}}}, \\
\left\langle\Gamma,\left(\mathfrak{C}-\mathrm{C}_{\mathcal{X}\mathcal{Y}}\otimes_{L(L(\mathcal{H}_{\mathcal{Y}},\mathcal{H}_{\mathcal{X}}))}\mathrm{C}_{\mathcal{X}\mathcal{Y}}\right)\Gamma\right\rangle_{L(\mathcal{H}_{\mathcal{Y}},\mathcal{H}_{\mathcal{X}})}\leq\tau^{-2}r\Big\}
\end{aligned}
$$

*is bounded as,*

$$
\mathfrak{R}_n(\mathcal{Q}_r) \leq \frac{1}{\sqrt{n}}\inf_{h\geq0}\left(\sqrt{r}h+2\tau\sqrt{\frac{k}{\lambda_{\mathcal{X}}\lambda_{\mathcal{Y}}}\sum_{j>h}\lambda_j(\mathfrak{C}')}\right)
$$

*Proof.* Let $\phi_i$'s be eigenfunctions of $\mathfrak{C}' = \mathfrak{C}-\mathrm{C}_{\mathcal{X}\mathcal{Y}}\otimes_{L(L(\mathcal{H}_{\mathcal{Y}},\mathcal{H}_{\mathcal{X}}))}\mathrm{C}_{\mathcal{X}\mathcal{Y}}$ which form an orthonormal basis. For $h\leq\mathrm{rank}(\mathfrak{C}')$, we have

$$
\begin{aligned}
&\sum_{i=1}^n\sigma_i\langle\Gamma,\mathrm{C}_{x_i,y_i}-\mathrm{C}_{\mathcal{X}\mathcal{Y}}\rangle_{L(\mathcal{H}_{\mathcal{Y}},\mathcal{H}_{\mathcal{X}})} = \sum_{j\geq1}\langle\Gamma,\phi_j\rangle_{L(\mathcal{H}_{\mathcal{Y}},\mathcal{H}_{\mathcal{X}})}\left\langle\phi_j,\sum_{i=1}^n\sigma_i(\mathrm{C}_{x_i,y_i}-\mathrm{C}_{\mathcal{X}\mathcal{Y}})\right\rangle_{L(\mathcal{H}_{\mathcal{Y}},\mathcal{H}_{\mathcal{X}})} \\
&= \sum_{j=1}^h\langle\Gamma,\phi_j\rangle_{L(\mathcal{H}_{\mathcal{Y}},\mathcal{H}_{\mathcal{X}})}\sqrt{\lambda_j(\mathfrak{C}')}\left\langle\phi_j,\sum_{i=1}^n\sigma_i(\mathrm{C}_{x_i,y_i}-\mathrm{C})\right\rangle_{L(\mathcal{H}_{\mathcal{Y}},\mathcal{H}_{\mathcal{X}})}\cdot\frac{1}{\sqrt{\lambda_j(\mathfrak{C}')}} \\
&\quad + \sum_{j>h}\langle\Gamma,\phi_j\rangle_{L(\mathcal{H}_{\mathcal{Y}},\mathcal{H}_{\mathcal{X}})}\left\langle\phi_j,\sum_{i=1}^n\sigma_i(\mathrm{C}_{x_i,y_i}-\mathrm{C}_{\mathcal{X}\mathcal{Y}})\right\rangle_{L(\mathcal{H}_{\mathcal{Y}},\mathcal{H}_{\mathcal{X}})} \\
&\leq \left(\sum_{j=1}^h\langle\Gamma,\phi_j\rangle^2_{L(\mathcal{H}_{\mathcal{Y}},\mathcal{H}_{\mathcal{X}})}\lambda_j(\mathfrak{C}')\right)^{1/2}\left(\sum_{j=1}^h\left\langle\phi_j,\sum_{i=1}^n\sigma_i(\mathrm{C}_{x_i,y_i}-\mathrm{C}_{\mathcal{X}\mathcal{Y}})\right\rangle^2_{L(\mathcal{H}_{\mathcal{Y}},\mathcal{H}_{\mathcal{X}})}\frac{1}{\lambda_j(\mathfrak{C}')}\right)^{1/2} \\
&\quad + \left(\sum_{j>h}\langle\Gamma,\phi_j\rangle^2_{L(\mathcal{H}_{\mathcal{Y}},\mathcal{H}_{\mathcal{X}})}\right)^{1/2}\left(\sum_{j>h}\left\langle\phi_j,\sum_{i=1}^n\sigma_i(\mathrm{C}_{x_i,y_i}-\mathrm{C}_{\mathcal{X}\mathcal{Y}})\right\rangle^2_{L(\mathcal{H}_{\mathcal{Y}},\mathcal{H}_{\mathcal{X}})}\right)^{1/2} \\
&\leq \frac{\sqrt{r}}{\tau}\left(\sum_{j=1}^h\frac{1}{\lambda_j(\mathfrak{C}')}\left\langle\phi_j,\sum_{i=1}^n\sigma_i(\mathrm{C}_{x_i,y_i}-\mathrm{C}_{\mathcal{X}\mathcal{Y}})\right\rangle^2_{L(\mathcal{H}_{\mathcal{Y}},\mathcal{H}_{\mathcal{X}})}\right)^{1/2} \\
&\quad + \left(\frac{4k}{\lambda_{\mathcal{X}}\lambda_{\mathcal{Y}}}\right)^{1/2}\left(\sum_{j>h}\left\langle\phi_j,\sum_{i=1}^n\sigma_i(\mathrm{C}_{x_i,y_i}-\mathrm{C}_{\mathcal{X}\mathcal{Y}})\right\rangle^2_{L(\mathcal{H}_{\mathcal{Y}},\mathcal{H}_{\mathcal{X}})}\right)^{1/2}
\end{aligned}
$$

where in the third step, we used Cauchy Schwartz inequality. In the fourth step, we use that

$$\|\Gamma\|^2_{L(\mathcal{H}_\mathcal{Y},\mathcal{H}_\mathcal{X})} = \sum_i \langle \Gamma, \phi_i \rangle^2_{L(\mathcal{H}_\mathcal{Y},\mathcal{H}_\mathcal{X})} \leq \frac{4k}{\lambda_\mathcal{X}\lambda_\mathcal{Y}}$$

and

$$
\begin{aligned}
\langle \Gamma, \mathfrak{C}'\Gamma \rangle_{L(\mathcal{H}_\mathcal{Y},\mathcal{H}_\mathcal{X})} &= \left\langle \Gamma, \left( \sum_i \lambda_i(\mathfrak{C}')\phi_i \otimes_{L(\mathcal{H}_\mathcal{Y},\mathcal{H}_\mathcal{X})} \phi_i \right) \Gamma \right\rangle_{L(\mathcal{H}_\mathcal{Y},\mathcal{H}_\mathcal{X})} \\
&= \sum_i \lambda_i(\mathfrak{C}') \left\langle \Gamma, \left( \phi_i \otimes_{L(\mathcal{H}_\mathcal{Y},\mathcal{H}_\mathcal{X})} \phi_i \right) \Gamma \right\rangle_{L(\mathcal{H}_\mathcal{Y},\mathcal{H}_\mathcal{X})} \\
&= \sum_i \lambda_i(\mathfrak{C}') \left\langle \Gamma, \left( \langle \Gamma, \phi_i \rangle_{L(\mathcal{H}_\mathcal{Y},\mathcal{H}_\mathcal{X})} \phi_i \right) \right\rangle_{L(\mathcal{H}_\mathcal{Y},\mathcal{H}_\mathcal{X})} \\
&= \sum_i \lambda_i(\mathfrak{C}') \langle \Gamma, \phi_i \rangle^2_{L(\mathcal{H}_\mathcal{Y},\mathcal{H}_\mathcal{X})} \leq \tau^{-2}r
\end{aligned}
$$

where the third equality follows from the definition of the outer product.
We now look at,

$$
\begin{aligned}
\mathbb{E}_{x,y,\sigma} \left[ \left\langle \sum_{j=1}^n \sigma_j \left( C_{x_j,y_j} - C_{\mathcal{X}\mathcal{Y}} \right), \phi_i \right\rangle^2_{L(\mathcal{H}_\mathcal{Y},\mathcal{H}_\mathcal{X})} \right] &= \mathbb{E}_{x_j,y_j,\sigma} \left[ \sum_{j=1}^n \sigma_j^2 \left\langle C_{x_j,y_j} - C_{\mathcal{X}\mathcal{Y}}, \phi_i \right\rangle^2_{L(\mathcal{H}_\mathcal{Y},\mathcal{H}_\mathcal{X})} \right] \\
&= \mathbb{E}_{x,y} \left[ \sum_{j=1}^n \left\langle C_{x_j,y_j} - C_{\mathcal{X}\mathcal{Y}}, \phi_i \right\rangle^2_{L(\mathcal{H}_\mathcal{Y},\mathcal{H}_\mathcal{X})} \right] \\
&= \mathbb{E}_{x,y} \left[ \left\langle \phi_i, \left( \sum_{j=1}^n \left( C_{x_j,y_j} - C_{\mathcal{X}\mathcal{Y}} \right) \otimes_{L(\mathcal{H}_\mathcal{Y},\mathcal{H}_\mathcal{X})} \left( C_{x_j,y_j} - C_{\mathcal{X}\mathcal{Y}} \right) \right) \phi_i \right\rangle_{L(\mathcal{H}_\mathcal{Y},\mathcal{H}_\mathcal{X})} \right] \\
&= \left\langle \phi_i, \mathbb{E}_{x,y} \left[ \sum_{j=1}^n \left( C_{x_j,y_j} - C \right) \otimes_{L(\mathcal{H}_\mathcal{Y},\mathcal{H}_\mathcal{X})} \left( C_{x_j,y_j} - C \right) \right] \phi_i \right\rangle_{L(\mathcal{H}_\mathcal{Y},\mathcal{H}_\mathcal{X})} \\
&= n \left\langle \phi_i, \mathfrak{C}'\phi_i \right\rangle_{L(\mathcal{H}_\mathcal{Y},\mathcal{H}_\mathcal{X})} \\
&= n\lambda_i(\mathfrak{C}')
\end{aligned}
$$

where in the first step, we use Pythagoras theorem by observing that $\phi_i$'s form an orthonormal basis; and in the second step, we use that fact that for a Rademacher variable $\sigma$, $\mathbb{E}\left[\sigma^2\right] = 1$. In the fourth step, we use that $\mathfrak{C}' = \mathbb{E}_{x,y}\left[ \frac{1}{n} \sum_{j=1}^n \left( C_{x_j,y_j} - C_{\mathcal{X}\mathcal{Y}} \right) \otimes_{L(\mathcal{H}_\mathcal{Y},\mathcal{H}_\mathcal{X})} \left( C_{x_j,y_j} - C_{\mathcal{X}\mathcal{Y}} \right) \right]$, and in the fifth step, we use the fact that $\phi_i$ is an eigenfunction of $\mathfrak{C}'$.

Let $\mathcal{G}_r$ denote the feasible set of $\Gamma$ defined as

$$\mathcal{G}_r := \left\{ \Gamma \in L(\mathcal{H}_\mathcal{Y},\mathcal{H}_\mathcal{X}) \mid \|\Gamma\|^2_{L(\mathcal{H}_\mathcal{Y},\mathcal{H}_\mathcal{X})} \leq \frac{4k}{\lambda_\mathcal{X}\lambda_\mathcal{Y}}, \langle \Gamma, \mathfrak{C}\Gamma \rangle_{L(\mathcal{H}_\mathcal{Y},\mathcal{H}_\mathcal{X})} \leq \tau^{-2}r \right\}$$

We have,

$$
\Re_m(\mathcal{Q}_r) = \frac{\tau}{n} \mathbb{E}_{x,y,\sigma} \left[ \sup_{f \in \mathcal{Q}_r} \sum_{i=1}^n \sigma_i f(x_i, y_i) \right]
$$

$$
= \frac{\tau}{n} \mathbb{E}_{x,y,\sigma} \left[ \sup_{\Gamma \in \mathcal{G}_r} \sum_{i=1}^n \sigma_i \left\langle \Gamma, \mathrm{C}_{x_i,y_i} - \mathrm{C}_{\mathcal{XY}} \right\rangle_{L(\mathcal{H}_\mathcal{Y}, \mathcal{H}_\mathcal{X})} \right]
$$

$$
\leq \frac{\tau}{n} \mathop{\mathbb{E}}_{x,y,\sigma} \left[ \sup_{\Gamma \in \mathcal{G}_r} \left( \frac{r}{\tau^2} \sum_{j=1}^h \frac{1}{\lambda_i(\mathfrak{C}')} \right)^{1/2} \left( \sum_{j=1}^h \left\langle \phi_j, \sum_{i=1}^n \sigma_i (\mathrm{C}_{x_i,y_i} - \mathrm{C}_{\mathcal{XY}}) \right\rangle^2_{L(\mathcal{H}_\mathcal{Y}, \mathcal{H}_\mathcal{X})} \right)^{1/2} \right.
$$

$$
\left. + \left( \frac{4k}{\lambda_\mathcal{X} \lambda_\mathcal{Y}} \right)^{1/2} \left( \sum_{j>h} \left\langle \phi_j, \sum_{i=1}^n \sigma_i (\mathrm{C}_{x_i,y_i} - \mathrm{C}_{\mathcal{XY}}) \right\rangle^2_{L(\mathcal{H}_\mathcal{Y}, \mathcal{H}_\mathcal{X})} \right)^{1/2} \right]
$$

$$
\leq \frac{\tau}{n} \left( \frac{\sqrt{r}}{\tau} \left( \sum_{j=1}^h \frac{1}{\lambda_j(\mathfrak{C}')} \mathbb{E}_{x,y,\sigma} \left[ \left\langle \phi_j, \sum_{i=1}^n \sigma_i (\mathrm{C}_{x_i,y_i} - \mathrm{C}_{\mathcal{XY}}) \right\rangle^2_{L(\mathcal{H}_\mathcal{Y}, \mathcal{H}_\mathcal{X})} \right] \right)^{1/2} \right.
$$

$$
\left. + \left( \frac{4k}{\lambda_\mathcal{X} \lambda_\mathcal{Y}} \right)^{1/2} \left( \sum_{j>h} \mathbb{E}_{x,y,\sigma} \left[ \left\langle \phi_j, \sum_{i=1}^n \sigma_i (\mathrm{C}_{x_i,y_i} - \mathrm{C}) \right\rangle^2_{L(\mathcal{H}_\mathcal{Y}, \mathcal{H}_\mathcal{X})} \right] \right)^{1/2} \right)
$$

$$
\leq \frac{1}{\sqrt{n}} \left( \sqrt{r} \sum_{j=1}^h \frac{\lambda_j(\mathfrak{C}')}{\lambda_j(\mathfrak{C}')} + \sqrt{\sum_{j>h} \lambda_j(\mathfrak{C}')} \frac{2\sqrt{k}}{\sqrt{\lambda_\mathcal{X} \lambda_\mathcal{Y}}} \right)
$$

$$
= \frac{1}{\sqrt{n}} \left( \sqrt{r}h + 2\tau \sqrt{\frac{k}{\lambda_\mathcal{X} \lambda_\mathcal{Y}} \sum_{j>h} \lambda_j(\mathfrak{C}')} \right)
$$

Since the above holds for all $h \leq \mathrm{rank}(\mathfrak{C}')$ and can be trivially extended to $h \geq \mathrm{rank}(\mathfrak{C}')$ as $\lambda_j(\mathfrak{C}') = 0$ for $j > \mathrm{rank}(\mathfrak{C}')$, it therefore holds for the infimum over $h$. We therefore have,

$$
\Re_n(\mathcal{Q}_r) \leq \frac{1}{\sqrt{n}} \inf_{h \geq 0} \left( \sqrt{r}h + 2\tau \sqrt{\frac{k}{\lambda_\mathcal{X} \lambda_\mathcal{Y}} \sum_{j>h} \lambda_j(\mathfrak{C}')} \right)
$$

$\square$

**Lemma 21.** *The fixed point of $\psi(r)$, i.e.*

$$
r^* = \psi(r^*) = \frac{\xi\tau}{\sqrt{n}} \left( \sqrt{r^*} \left( \sqrt{h} + 1 \right) + 2\tau \sqrt{\frac{k}{\lambda_\mathcal{X} \lambda_\mathcal{Y}} \sum_{j>h} \lambda_j(\mathfrak{C}')} \right)
$$

*is bounded as,*

$$
r^* \leq \tau^2 \left( \inf_{h \geq 0} \left\{ \frac{\xi^2 h}{n} + \frac{2\xi^2 \sqrt{h}}{n} + \frac{4\xi}{\sqrt{m}} \sqrt{\frac{k}{\lambda_\mathcal{X} \lambda_\mathcal{Y}} \sum_{j>h} \lambda_j(\mathfrak{C}')} \right\} + \frac{\xi^2}{n} \right)
$$

*Proof.* We have,

$$
r^* = \psi(r^*) = \frac{\xi\tau}{\sqrt{n}} \left( \sqrt{r^*} \left( \sqrt{h} + 1 \right) + 2\tau \sqrt{\frac{k}{\lambda_\mathcal{X} \lambda_\mathcal{Y}} \sum_{j>h} \lambda_j(\mathfrak{C}')} \right)
$$

Consider the quadratic equation $x - a\sqrt{x} - b \leq 0$, we have,

$$x \leq \left( \frac{a + \sqrt{a^2 + 4b}}{2} \right)^2$$

$$\leq \frac{2a^2 + 4b + 2\sqrt{a^2(a^2 + 4b)}}{4}$$

$$\leq a^2 + 2b$$

where in the last step, we use that geometric mean $\leq$ arithmetic mean. Plugging it here, we get

$$r^* \leq \frac{\xi^2 \tau^2}{n} \left( h + 1 + 2\sqrt{h} \right) + \frac{4\xi \tau^2}{\sqrt{n}} \sqrt{\frac{k}{\lambda_{\mathcal{X}} \lambda_{\mathcal{Y}}} \sum_{j>h} \lambda_j(\mathfrak{C}')}$$

Taking infimum over $h$, we get,

$$r^* \leq \tau^2 \left( \inf_{h \geq 0} \left\{ \frac{\xi^2 h}{n} + \frac{2\xi^2 \sqrt{h}}{n} + \frac{4\xi}{\sqrt{n}} \sqrt{\frac{k}{\lambda_{\mathcal{X}} \lambda_{\mathcal{Y}}} \sum_{j>h} \lambda_j(\mathfrak{C}')} \right\} + \frac{\xi^2}{n} \right)$$

$\square$

**Lemma 22.** *For any $f \in \mathcal{G}$, $\mathbb{E}\left[f^2\right] \leq \mu \mathbb{E}\left[f\right]$ where $\mu = \frac{2\alpha_\rho \tau}{\lambda_{\mathcal{X}} \lambda_{\mathcal{Y}}}$ and $\alpha_\rho = \frac{\mathbb{E}_{x,y,x',y'}\left[ \langle x,x' \rangle_{\mathcal{H}_{\mathcal{X}}}^2 \langle y,y' \rangle_{\mathcal{H}_{\mathcal{Y}}}^2 \right]}{(\lambda_k(C) - \lambda_{k+1}(C))}$.*

*Proof.* Given $U, V$, define $\overline{U} := \left( C_{\mathcal{X}}^{\lambda_{\mathcal{X}}} \right)^{1/2} U, \bar{V} := \left( C_{\mathcal{Y}}^{\lambda_{\mathcal{Y}}} \right)^{1/2} V$. We remind that $U_{C,\lambda} = \left( C_{\mathcal{X}}^{\lambda_{\mathcal{X}}} \right)^{1/2} \overline{U_{C,\lambda}}$ and $V_{C,\lambda} = \left( C_{\mathcal{X}}^{\lambda_{\mathcal{X}}} \right)^{1/2} \overline{V_{C,\lambda}}$, so we get $U_{C,\lambda}^* U_{C,\lambda} = I$ and $V_{C,\lambda}^* V_{C,\lambda} = I$. Define the projection $P_{C,\lambda} := U_{C,\lambda} V_{C,\lambda}^* = \sum_{i=1}^k u_i^{C_\lambda} \otimes_{L(\mathcal{H}_{\mathcal{Y}}, \mathcal{H}_{\mathcal{X}})} v_i^{C_\lambda}$ and $P := \overline{U}\overline{V}^* = \sum_{i=1}^k u_i \otimes_{L(\mathcal{H}_{\mathcal{Y}}, \mathcal{H}_{\mathcal{X}})} v_i$ using their singular value decomposition respectively. Let $\bar{f} \in \mathcal{G}$. We first look at $\mathbb{E}\left[\bar{f}^2\right]$.

$$\mathbb{E}\left[\bar{f}^2\right] = \mathbb{E}\left[ \left\langle \overline{U}_{C,\lambda} \overline{V}_{C,\lambda}^* - UV^*, C_{x,y} \right\rangle_{L(\mathcal{H}_{\mathcal{Y}}, \mathcal{H}_{\mathcal{X}})}^2 \right]$$

$$= \left\langle \overline{U}_{C,\lambda} \overline{V}_{C,\lambda}^* - UV^*, \mathbb{E}\left[ (C_{x,y} \otimes_{L(\mathcal{H}_{\mathcal{Y}}, \mathcal{H}_{\mathcal{X}})} C_{x,y}) \right] (\overline{U}_{C,\lambda} \overline{V}_{C,\lambda}^* - UV^*) \right\rangle_{L(\mathcal{H}_{\mathcal{Y}}, \mathcal{H}_{\mathcal{X}})}$$

$$= \left\langle \overline{U}_{C,\lambda} \overline{V}_{C,\lambda}^* - UV^*, \mathfrak{C}(\overline{U}_{C,\lambda} \overline{V}_{C,\lambda}^* - UV^*) \right\rangle_{L(\mathcal{H}_{\mathcal{Y}}, \mathcal{H}_{\mathcal{X}})}$$

$$= \|\mathfrak{C}\|_{L(L(\mathcal{H}_{\mathcal{Y}}, \mathcal{H}_{\mathcal{X}}))} \left\| \overline{U}_{C,\lambda} \overline{V}_{C,\lambda}^* - UV^* \right\|_{L(\mathcal{H}_{\mathcal{Y}}, \mathcal{H}_{\mathcal{X}})}^2$$

$$= \|\mathfrak{C}\|_{L(L(\mathcal{H}_{\mathcal{Y}}, \mathcal{H}_{\mathcal{X}}))} \left\| \left( C_{\mathcal{X}}^{\lambda_{\mathcal{X}}} \right)^{-1/2} \left( U_{C,\lambda} V_{C,\lambda}^* - \overline{UV}^* \right) C_{\mathcal{Y}}^{\lambda_{\mathcal{Y}} -1/2} \right\|_{L(\mathcal{H}_{\mathcal{Y}}, \mathcal{H}_{\mathcal{X}})}^2$$

$$= \|\mathfrak{C}\|_{L(L(\mathcal{H}_{\mathcal{Y}}, \mathcal{H}_{\mathcal{X}}))} \left\| \left( C_{\mathcal{X}}^{\lambda_{\mathcal{X}}} \right)^{-1/2} \left( P_{C,\lambda} - P \right) \left( C_{\mathcal{Y}}^{\lambda_{\mathcal{Y}}} \right)^{-1/2} \right\|_{L(\mathcal{H}_{\mathcal{Y}}, \mathcal{H}_{\mathcal{X}})}^2$$

$$\leq \frac{1}{\lambda_{\mathcal{X}} \lambda_{\mathcal{Y}}} \|\mathfrak{C}\|_{L(L(\mathcal{H}_{\mathcal{Y}}, \mathcal{H}_{\mathcal{X}}))} \|P_{C,\lambda} - P\|_{L(\mathcal{H}_{\mathcal{Y}}, \mathcal{H}_{\mathcal{X}})}^2$$

$$= \frac{2}{\lambda_{\mathcal{X}} \lambda_{\mathcal{Y}}} \|\mathfrak{C}\|_{L(L(\mathcal{H}_{\mathcal{Y}}, \mathcal{H}_{\mathcal{X}}))} \left( k - \langle P_{C,\lambda}, P \rangle_{L(\mathcal{H}_{\mathcal{Y}}, \mathcal{H}_{\mathcal{X}})} \right)$$

$$= \frac{2}{\lambda_{\mathcal{X}} \lambda_{\mathcal{Y}}} \|\mathfrak{C}\|_{L(L(\mathcal{H}_{\mathcal{Y}}, \mathcal{H}_{\mathcal{X}}))} \left( k - \sum_{i,j=1}^k \left\langle u_i^{C_\lambda}, u_j \right\rangle_{\mathcal{H}_{\mathcal{X}}} \left\langle v_i^{C_\lambda}, v_j \right\rangle_{\mathcal{H}_{\mathcal{Y}}} \right)$$

where in the seventh inequality we just expanded $\|\mathrm{P}_{\mathrm{C},\lambda} - \mathrm{P}\|^2_{L(\mathcal{H}_\mathcal{Y},\mathcal{H}_\mathcal{X})}$. Now, note that

$$
\begin{aligned}
\|\mathfrak{C}\|^2_{L(L(\mathcal{H}_\mathcal{Y},\mathcal{H}_\mathcal{X}))} &= \langle \mathfrak{C}, \mathfrak{C} \rangle_{L(L(\mathcal{H}_\mathcal{Y},\mathcal{H}_\mathcal{X}))} \\
&= \mathbb{E}_{\mathrm{x,y,x',y'}} \left[ \left\langle \mathrm{C_{x,y}} \otimes_{L(\mathcal{H}_\mathcal{Y},\mathcal{H}_\mathcal{X})} \mathrm{C_{x,y}}, \mathrm{C_{x',y'}} \otimes_{L(\mathcal{H}_\mathcal{Y},\mathcal{H}_\mathcal{X})} \mathrm{C_{x',y'}} \right\rangle_{L(L(\mathcal{H}_\mathcal{Y},\mathcal{H}_\mathcal{X}))} \right] \\
&= \mathbb{E}_{\mathrm{x,y,x',y'}} \left[ \langle \mathrm{C_{x,y}}, \mathrm{C_{x',y'}} \rangle^2_{L(\mathcal{H}_\mathcal{Y},\mathcal{H}_\mathcal{X})} \right] \\
&= \mathbb{E}_{\mathrm{x,y,x',y'}} \left[ \langle \mathrm{x} \otimes \mathrm{y}, \mathrm{x'} \otimes \mathrm{y'} \rangle^2_{L(\mathcal{H}_\mathcal{Y},\mathcal{H}_\mathcal{X})} \right] \\
&= \mathbb{E}_{\mathrm{x,y,x',y'}} \left[ \langle \mathrm{x}, \mathrm{x'} \rangle^2_{\mathcal{H}_\mathcal{X}} \langle \mathrm{y}, \mathrm{y'} \rangle^2_{\mathcal{H}_\mathcal{Y}} \right] =: \alpha_\rho
\end{aligned}
$$

Let us now look at $\mathbb{E}\left[\bar{f}\right]$.

$$
\begin{aligned}
\mathbb{E}\left[\bar{f}\right] &= \mathbb{E}\left[ \left\langle \overline{\mathrm{U}}_{\mathrm{C},\lambda} \overline{\mathrm{V}}^*_{\mathrm{C},\lambda} - \mathrm{UV}^*, \mathrm{C_{x,y}} \right\rangle_{L(\mathcal{H}_\mathcal{Y},\mathcal{H}_\mathcal{X})} \right] \\
&= \left\langle \overline{\mathrm{U}}_{\mathrm{C},\lambda} \overline{\mathrm{V}}^*_{\mathrm{C},\lambda} - \mathrm{UV}^*, \mathrm{C}_{\mathcal{X}\mathcal{Y}} \right\rangle_{L(\mathcal{H}_\mathcal{Y},\mathcal{H}_\mathcal{X})} \\
&= \left\langle \mathrm{P}_{\mathrm{C},\lambda} - \mathrm{P}, \left(\mathrm{C}^{\lambda_\mathcal{X}}_\mathcal{X}\right)^{-1/2} \mathrm{C}_{\mathcal{X}\mathcal{Y}} \left(\mathrm{C}^{\lambda_\mathcal{Y}}_\mathcal{Y}\right)^{-1/2} \right\rangle_{L(\mathcal{H}_\mathcal{Y},\mathcal{H}_\mathcal{X})} \\
&= \langle \mathrm{P}_{\mathrm{C},\lambda} - \mathrm{P}, \mathrm{C}_\lambda \rangle_{L(\mathcal{H}_\mathcal{Y},\mathcal{H}_\mathcal{X})} \\
&= \sum_{i=1}^k \left( \sigma_i(\mathrm{C}_\lambda) - \langle u_i, \mathrm{C}_\lambda v_i \rangle_{\mathcal{H}_\mathcal{X}} \right)
\end{aligned}
$$

Let $u_i = \sum_{j=1}^k \left\langle u_i, u_j^{\mathrm{C}_\lambda} \right\rangle_{\mathcal{H}_\mathcal{X}} u_j^{\mathrm{C}_\lambda} + r_i$, where $r_i$ is orthogonal to $u_j^{\mathrm{C}_\lambda}, j \in [k]$ and $v_i = \sum_{j=1}^k \left\langle v_i, v_j^{\mathrm{C}_\lambda} \right\rangle_{\mathcal{H}_\mathcal{Y}} v_j^{\mathrm{C}_\lambda} + s_i$, where $s_i$ is orthogonal to $v_j^{\mathrm{C}_\lambda}, j \in [k]$. Then

$$
\begin{aligned}
\langle u_i, \mathrm{C}_\lambda v_i \rangle_{\mathcal{H}_\mathcal{X}} &= \left\langle \sum_{j=1}^k \left\langle u_i, u_j^{\mathrm{C}_\lambda} \right\rangle_{\mathcal{H}_\mathcal{X}} u_j^{\mathrm{C}_\lambda} + r_i, \mathrm{C}_\lambda \left( \sum_{j=1}^k \left\langle v_i, v_j^{\mathrm{C}_\lambda} \right\rangle_{\mathcal{H}_\mathcal{Y}} v_j^{\mathrm{C}_\lambda} + s_i \right) \right\rangle \\
&= \left\langle \sum_{j=1}^k \left\langle u_i, u_j^{\mathrm{C}_\lambda} \right\rangle_{\mathcal{H}_\mathcal{X}} u_j^{\mathrm{C}_\lambda} + r_i, \sum_{j=1}^k \lambda_j(\mathrm{C}_\lambda) \left\langle v_i, v_j^{\mathrm{C}_\lambda} \right\rangle_{\mathcal{H}_\mathcal{Y}} u_j^{\mathrm{C}_\lambda} + \mathrm{C}_\lambda s_i \right\rangle \\
&= \sum_{j=1}^k \lambda_j(\mathrm{C}_\lambda) \left\langle u_i, u_j^{\mathrm{C}_\lambda} \right\rangle_{\mathcal{H}_\mathcal{X}} \left\langle v_i, v_j^{\mathrm{C}_\lambda} \right\rangle_{\mathcal{H}_\mathcal{Y}} + \langle r_i, \mathrm{C}_\lambda s_i \rangle_{\mathcal{H}_\mathcal{X}}
\end{aligned}
$$

The cross terms are zero because $C_\lambda s_i$'s will be a linear combination of $u_i, i > k$ and so orthogonal to $u_j, j \in [k]$. Note that

$$
\begin{aligned}
\langle r_i, C_\lambda s_i \rangle_{\mathcal{H}_\mathcal{X}} &\leq \lambda_{k+1}(C_\lambda) \|r_i\|_{\mathcal{H}_\mathcal{X}} \|s_i\|_{\mathcal{H}_\mathcal{Y}} \\
&= \lambda_{k+1}(C_\lambda) \left\| u_i - \sum_{j=1}^{k} \left\langle u_i, u_j^{C_\lambda} \right\rangle u_j^{C_\lambda} \right\|_{\mathcal{H}_\mathcal{X}} \cdot \left\| v_i - \sum_{j=1}^{k} \left\langle v_i, v_j^{C_\lambda} \right\rangle v_j^{C_\lambda} \right\|_{\mathcal{H}_\mathcal{Y}} \\
&= \lambda_{k+1}(C_\lambda) \sqrt{1 - \sum_{j=1}^{k} \left\langle u_i, u_j^{C_\lambda} \right\rangle_{\mathcal{H}_\mathcal{X}}^2} \sqrt{1 - \sum_{j=1}^{k} \left\langle v_i, v_j^{C_\lambda} \right\rangle_{\mathcal{H}_\mathcal{Y}}^2} \\
&\leq \lambda_{k+1}(C_\lambda) \left( 1 - \frac{\sum_{j=1}^{k} \left\langle u_i, u_j^{C_\lambda} \right\rangle_{\mathcal{H}_\mathcal{X}}^2 + \sum_{j=1}^{k} \left\langle v_i, v_j^{C_\lambda} \right\rangle_{\mathcal{H}_\mathcal{Y}}^2}{2} \right) \\
&\leq \lambda_{k+1}(C_\lambda) \left( 1 - \sqrt{\left( \sum_{j=1}^{k} \left\langle u_i, u_j^{C_\lambda} \right\rangle_{\mathcal{H}_\mathcal{X}}^2 \right) \left( \sum_{j=1}^{k} \left\langle v_i, v_j^{C_\lambda} \right\rangle_{\mathcal{H}_\mathcal{Y}}^2 \right)} \right) \\
&\leq \lambda_{k+1}(C_\lambda) \left( 1 - \sum_{j=1}^{k} \left\langle u_i, u_j^{C_\lambda} \right\rangle_{\mathcal{H}_\mathcal{X}} \left\langle v_i, v_j^{C_\lambda} \right\rangle_{\mathcal{H}_\mathcal{Y}} \right)
\end{aligned}
$$

where the first step follows because $r_i$ and $s_i$ don't include components along the first $k$ $u_i$'s and $v_i$ respectively. In the fourth and fifth steps, we use that arithmetic mean $\geq$ geometric mean, and in the last step, we use Cauchy-Schwartz inequality. Plugging this in the previous bound, we get,

$$
\langle u_i, C_\lambda v_i \rangle_{\mathcal{H}_\mathcal{X}} \leq \sum_{j=1}^{k} \lambda_j(C_\lambda) \left\langle u_i, u_j^{C_\lambda} \right\rangle_{\mathcal{H}_\mathcal{X}} \left\langle v_i, v_j^{C_\lambda} \right\rangle_{\mathcal{H}_\mathcal{Y}} + \lambda_{k+1}(C_\lambda) \left( 1 - \sum_{j=1}^{k} \left\langle u_i, u_j^{C_\lambda} \right\rangle_{\mathcal{H}_\mathcal{X}} \left\langle v_i, v_j^{C_\lambda} \right\rangle_{\mathcal{H}_\mathcal{Y}} \right)
$$

Moreover,

$$
\begin{aligned}
\mathbb{E}\left[\bar{f}\right] &\geq \sum_{i=1}^{k} \left( \lambda_i(C_\lambda) - \left( \sum_{j=1}^{k} \lambda_j(C_\lambda) \left\langle u_i, u_j^{C_\lambda} \right\rangle_{\mathcal{H}_\mathcal{X}} \left\langle v_i, v_j^{C_\lambda} \right\rangle_{\mathcal{H}_\mathcal{Y}} + \lambda_{k+1}(C_\lambda) \left( 1 - \sum_{j=1}^{k} \left\langle u_i, u_j^{C_\lambda} \right\rangle_{\mathcal{H}_\mathcal{X}} \left\langle v_i, v_j^{C} \right\rangle_{\mathcal{H}_\mathcal{Y}} \right) \right) \right) \\
&= \sum_{i=1}^{k} \lambda_i(C_\lambda) \left( 1 - \sum_{j=1}^{k} \left\langle u_i, u_j^{C_\lambda} \right\rangle_{\mathcal{H}_\mathcal{X}} \left\langle v_i, v_j^{C_\lambda} \right\rangle_{\mathcal{H}_\mathcal{Y}} \right) - \sum_{i=1}^{k} \lambda_{k+1}(C_\lambda) \left( 1 - \sum_{j=1}^{k} \left\langle u_i, u_j^{C_\lambda} \right\rangle_{\mathcal{H}_\mathcal{X}} \left\langle v_i, v_j^{C_\lambda} \right\rangle_{\mathcal{H}_\mathcal{Y}} \right) \\
&\geq k\left(\lambda_k(C_\lambda) - \lambda_{k+1}(C_\lambda)\right) - \left(\lambda_k(C_\lambda) - \lambda_{k+1}(C_\lambda)\right) \sum_{i,j=1}^{k} \left\langle u_i, u_j^{C_\lambda} \right\rangle_{\mathcal{H}_\mathcal{X}} \left\langle v_i, v_j^{C_\lambda} \right\rangle_{\mathcal{H}_\mathcal{Y}} \\
&= \left(\lambda_k(C_\lambda) - \lambda_{k+1}(C_\lambda)\right) \left( k - \sum_{i,j=1}^{k} \left\langle u_i, u_j^{C_\lambda} \right\rangle_{\mathcal{H}_\mathcal{X}} \left\langle v_i, v_j^{C_\lambda} \right\rangle_{\mathcal{H}_\mathcal{Y}} \right)
\end{aligned}
$$

where in the second last step, we used that $\lambda_i \geq \lambda_k, i \in [k]$ and $\left( 1 - \sum_{i,j=1}^{k} \left\langle u_i, u_j^{C_\lambda} \right\rangle_{\mathcal{H}_\mathcal{X}} \left\langle v_i, v_j^{C_\lambda} \right\rangle_{\mathcal{H}_\mathcal{Y}} \right) \geq \|r_i\|_{\mathcal{H}_\mathcal{X}} \|s_i\|_{\mathcal{H}_\mathcal{Y}} \geq 0$ (see above). Therefore, we get,

$$
\begin{aligned}
\mathbb{E}\left[\bar{f}\right] &\geq \left(\lambda_k(C_\lambda) - \lambda_{k+1}(C_\lambda)\right) \left( k - \sum_{i,j=1}^{k} \left\langle u_i, u_j^{C_\lambda} \right\rangle_{\mathcal{H}_\mathcal{X}} \left\langle v_i, v_j^{C_\lambda} \right\rangle_{\mathcal{H}_\mathcal{Y}} \right) \\
&\geq \frac{\left(\lambda_k(C_\lambda) - \lambda_{k+1}(C_\lambda)\right) \lambda_\mathcal{X} \lambda_\mathcal{Y}}{2\alpha_\rho} \mathbb{E}\left[\bar{f}^2\right]
\end{aligned}
$$

Let $\xi = \frac{2\alpha_\rho}{\lambda_{\mathcal{X}}\lambda_{\mathcal{Y}}}$. For $f \in \mathcal{G}$, let $\bar{f} = \tau^{-1} f$ where $\bar{f} \in \mathcal{F}$. Therefore,

$$
\begin{aligned}
\mathbb{E}\left[f^2\right] &= \tau^2 \mathbb{E}\left[\bar{f}^2\right] \\
&\leq \xi\tau^2 \mathbb{E}\left[\bar{f}\right] \\
&= \xi\tau\mathbb{E}\left[f\right] = \mu\mathbb{E}\left[f\right]
\end{aligned}
$$

where $\mu = \frac{2\alpha_\rho\tau}{\lambda_{\mathcal{X}}\lambda_{\mathcal{Y}}}$.

$\square$

