# OpenReview forum: "Generalization bounds for Kernel Canonical Correlation Analysis"
_TMLR — Accepted by TMLR_

### Review · Reviewer_mSDT · 2022-12-19

**Summary Of Contributions:**

This paper establishes non-asymptotic generalization error bounds for Kernel Canonical Correlation Analysis (KCCA).
Specifically, the bounds are given for the convergence of the objective value, rather than in terms of the estimator which is the standard practice in the literature.


**Audience:**

Yes

**Claims And Evidence:**

Yes

**Requested Changes:**

- CCA is mostly used as an intermediate step to other tasks and therefore it's really not clear what insights we can get by knowing the convergence of the loss.

- There seems to be an error in the proof of Lemma 13. Shouldn't the inequality (line 17, page 17) be CX^{1/2} u = CX^{−1/2}v - (CX^{−1/2}v − CX^{1/2}u)?

Typos:
- Lemma 13: there -> the
- 1 is missing in the inequality in Line 11, page 17


**Strengths And Weaknesses:**

The paper provides non-asymptotic bounds in terms of convergence in the objective for a well-established estimator. The authors motivated reasonably well the choice of focusing on convergence of the loss rather than of the estimators. Since the paper focuses on a different problem,  the comparison with the state-of-the-art rates may not be completely fair.

The analysis does not significantly improve upon previous results, and aims to control the excess risk of the CCA estimator rather than the distance between the estimator and the real solution.  Thus, it does not provide new perspectives that could improve practical approaches. In this sense, the paper has low significance and potential impact.

---

### Review · Reviewer_wHob · 2022-12-22

**Summary Of Contributions:**

In this paper, the authors establish (in Theorem 2) non-asymptotic bounds on the generalization error (in the sense of the objective) for regularized empirical risk minimizer of kernel canonical correlation analysis (KCCA). Results on the more general functional CCA are also given (in Theorem 1), i.e., when the Hilbert spaces of interest are not necessarily RKHS. The results also allow one to obtain guarantees on finite-dimensional (K)CCA (in Corollary 3) that improve previous results of linear CCA.

The technical results rely on a fine-grained decomposition of the excess error and local Rademacher complexity analysis.

**Audience:**

Yes

**Broader Impact Concerns:**

This contribution of this paper is mainly theoretical and I do not see any ethical concern to report.

**Claims And Evidence:**

Yes

**Requested Changes:**

Some detailed comments:

* P2 Sec 1.2 Our Contributions: depending on distributional properties OF THE DATA?
* P3: as "high confidence" -> as ``high confidence''
* The authors said at the end of P4 that "Observe that the solution to the CCA problem in Eqn. (1) is given by the singular value decomposition (SVD) of C" and then define U_C and V_C, would the authors make a more precise statement on how they relate to the solution of CCA problem? This is a basic conclusion, but of sufficient significance to be mentioned in the paper I believe.
* End of P5 "The regularization above corresponds to shifting the spectrum of the auto-covariance operators so that all eigenvalues are positive": for lambda_{\mathcal X}, lambda_{\mathcal Y} > 0?
* Section 3.1 in P5: perhaps add a few references here just to provide some context for applying Tikhonov regularization in CCA.
* Beginning of P6 "as is standard in statistical machine learning": perhaps cite a few papers on (optimal or not) n-dependent regularization in a, e.g., kernel ridge regression context.
* Assumption 1: perhaps point to the defining equation (1) of the operator C somewhere in Assumption 1.
* Does Theorem 1 need Assumption 1? If yes, this should be stated explicitly IN the theorem statement. Also, Assumption 1 assumes that r.v. x and y are bounded, but Theorem 1 only needs some almost sure boundedness. I feel confused about this.
* Theorem 1: some explanation is needed here in Theorem 1 on the parameter alpha_\rho, what are x' and y'? May independent copies of x,y?
* It would be great to have a table somewhere to summarize previous efforts and to better position this piece of contribution in the existing literature, e.g., to compare with the results in Fukumizu et al., 2007; Fan and Lian, 2016; Gao et al. (2017), etc.
* The authors argued in P7 that the proposed analysis is stronger than those in Fan and Lian (2016), in the sense that the proposed results cover the case of exponentially decaying eigenvalues. It would be of interest to discuss what novel insight can be obtained with the proposed analysis in this case. For the moment this remains very unclear.
* End of P7 "so both decrease with n when lambda = omega (n^{-1/2})": is this a typo?

**Strengths And Weaknesses:**

**Strengths**: This paper focuses on the important problem of the generalization error analysis of KCCA. And the proposed analysis improves previous arts in some sense.

**Weaknesses**: the presentation of the paper can clearly be improved. Efforts can be made to carefully revise the paper to help readers (myself including one of them) better understand the significance of this work and to position this piece of contribution in the existing literature. Some clarifications are needed, see below.

---

### Review · Reviewer_fTVa · 2023-01-16

**Summary Of Contributions:**


This paper studies the problem of  Canonical Correlation Analysis where the goal is, given two vectors X and Y, to find linear combinations of X and Y which have maximum correlation with each other. This problem is well-studied problem in Euclidean case. The main focus of the paper is on obtaining a non-asymptotic excess error guarantee for the case kernelized version of CCA. Also, the authors extend the result to the abstract Hilbert space.

The algorithm they consider is regularized ERM where we plug the empirical covariance matrices instead of the true ones.



**Audience:**

Yes

**Broader Impact Concerns:**

not applicable.

**Claims And Evidence:**

No

**Requested Changes:**

My main request is to improve the organization of the paper. One concrete suggestion is to focus on one instance of kernels for instance Gaussian kernels and discuss the results for this special case. Also, the motivation of the paper should be stated more clearly.

**Strengths And Weaknesses:**

- The motivation of the paper is not clear. The introduction does not convince me that studying Kernel CCA and extending the result to abstract Hilbert spaces is interesting.

- The presentation of the paper is not good. It is almost impossible to follow the paper. In particular, there are lots of notations everywhere in the paper. The relationship between the sections are not clear. For instance in Section 3, we have some subsections on ERM, Generalization error, and then Kernel Duality. It is not clear the connection of these parts.

- Do we have a proof that ERM without regularization fails for CCA problem?

- There are numerous assumptions in Section 3 without discussing the implication and intuitions of the these assumptions. I would suggest that the author should consider some case studies for their general statement.

- In Page 4 of the paper, the authors state many assumption. Then, suddenly before presenting the main result in Theorem 1, the authors state another assumption. I think the organization of the paper can be greatly improved.

- What is C in assumption 1?

- In the statistical learning theory, generalization error usually refers to the difference between the performance of a model on the test data and that of the training data. I am not sure you use the right terminology in the paper. I think in this paper the goal is to analyze "excess error".

- numerous typos in the appendix: Appendix A.1 first line has typos. Also there are other typos.  In Theorem 10, you use the probability notations that are not defined anywhere in the paper such as writing the expectation of f(X) under X~P as P[f]. Also, in Theorem 10 there are other typos as well.

- Statement of Theorem 11 in the appendix which is the main result of the paper can be improved. It is almost impossible to parse this theorem. The statement is more than 10 lines.

---

### Decision · Action_Editors · 2023-02-26

**Recommendation:** Accept as is

**Comment:**

The paper provides new results on kernel CCA which would be of interest to some in the ML community. The presentation is suitable.

**Audience:**

The paper provides non-asymptotic bounds for the convergence of the kernel CCA. The convergence is given in terms of the loss function as opposed to the resulting estimator. This differs from standard convergence results for the kernel CCA and could be interesting to certain TMLR readers.

**Claims And Evidence:**

After some revisions by the authors following reviewer comments, the paper now explains and substantiates its claims well.